# The amygdala is not necessary for the familiarity aspect of recognition memory

Benjamin M. Basile [1,2] ✉, Vincent D. Costa [1,3], Jamie L. Schafroth[1,4], Chloe L. Karaskiewicz[1,5], Daniel R. Lucas [1] & Elisabeth A. Murray [1]

Dual-process accounts of item recognition posit two memory processes: slow but detailed recollection, and quick but vague familiarity. It has been proposed, based on prior rodent work, that the amygdala is critical for the familiarity aspect of item recognition. Here, we evaluated this proposal in male rhesus monkeys (*Macaca mulatta*) with selective bilateral excitotoxic amygdala damage. We used four established visual memory tests designed to assess different aspects of familiarity, all administered on touchscreen computers. Specifically, we assessed monkeys' tendencies to make low-latency false alarms, to make false alarms to recently seen lures, to produce curvilinear ROC curves, and to discriminate stimuli based on repetition across days. Three of the four tests showed no familiarity impairment and the fourth was explained by a deficit in reward processing. Consistent with this, amygdala damage did produce an anticipated deficit in reward processing in a three-arm-bandit gambling task, verifying the effectiveness of the lesions. Together, these results contradict prior rodent work and suggest that the amygdala is not critical for the familiarity aspect of item recognition.

The embarrassing experience of recognizing a colleague at a conference as familiar but being unable to recollect their name is common and demonstrates the everyday dissociation between familiarity and recollection. The dominant theory of recognition posits two processes: a quick but vague familiarity process and a slow but detailed recollection process[1]. Although there are certainly disagreements about how these processes function, e.g., refs. 2,3, the balance of evidence supports the theory that recollection and familiarity are functionally separable[1].

One of the major undertakings of neuropsychology over the last few decades has been to identify the neural substrates of recollection and familiarity. Relatively more evidence exists about the brain areas supporting recollection, largely due to the greater prevalence of patients who suffer from selective recollection deficits, e.g., refs. 4–6. Much of this evidence about recollection suggests that the hippocampus is critical[4,7–9], though strong evidence also implicates extra-

hippocampal areas[6], e.g., refs. 10–12, and some evidence suggests no role for the hippocampus in visual recognition at all[13–15]. Relatively less evidence exists about the brain areas supporting familiarity, largely because few patients seem to demonstrate selective familiarity deficits.

The search for the neural substrates of familiarity has produced at least three major competing hypotheses. First, the initial report of a patient with a selective familiarity deficit claims that her impairment is most likely caused by damage to the perirhinal cortex[16]. Patient NB has a unilateral resection of her temporal lobe, resulting in 83% damage to her left amygdala, 59% damage to her left entorhinal cortex, and 43% damage to her left perirhinal cortex. As a result, she showed a robust deficit in familiarity memory across multiple paradigms, despite normal recognition accuracy. In the remember-know procedure, she showed reduced estimates of familiarity, largely resulting from increased false-alarms to items she said she just knew were vaguely

[1]Laboratory of Neuropsychology, National Institute of Mental Health, National Institutes of Health, Bethesda, MD 20892, USA. [2]Present address: Department of Psychology, Dickinson College, Carlisle, PA, USA. [3]Present address: Division of Neuroscience, Oregon National Primate Research Center, Portland, OR, USA. [4]Present address: School of Anthropology, University of Arizona, Tucson, AZ, USA. [5]Present address: Department of Psychology, UC Davis, Davis, CA, USA. ✉e-mail: basileb@dickinson.edu

familiar. With receiver operating characteristic (ROC) curves, in which hits and false alarms for studied targets and unstudied lures are plotted for different confidence judgments, her ROC curves were flattened relative to those of controls. The asymmetry in ROC curves has been proposed to measure the contribution of recollection, whereas the curvilinearity has been proposed to measure familiarity; thus, the flattening of NB's curve indicates an impairment in familiarity. Lastly, with a short response deadline, in which all subjects must rely on the faster familiarity process, NB's recognition accuracy was significantly impaired relative to a condition with no response deadline. The combination of increased false alarms for items she finds vaguely familiar, flattened ROC curves, and being disproportionately affected by a response deadline provided strong converging evidence for an impairment in familiarity. Despite being the structure with the least damage, the authors proposed that NB's perirhinal cortex damage was responsible for her familiarity impairment, largely based on prior correlational imaging work tying the perirhinal cortex to familiarity reviewed in ref. 17.

Second, a report of another patient with a selective familiarity deficit suggested that the critical structure is the entorhinal cortex[18]. Patient MR has a history of seizures and a small cavernoma that appears limited to her left entorhinal cortex. MR showed relatively normal overall recognition and estimates of recollection. However, on the remember-know procedure, she was significantly more likely than controls to make false alarms for items she said she knew were vaguely familiar. This increase in false alarms is strikingly similar to that seen with patient NB[16]. However, because MR's cavernoma seemed to only affect the entorhinal cortex, which was also damaged in patient NB, the authors proposed that the entorhinal cortex was the critical area for the familiarity aspect of recognition.

Third, a study of rodents has claimed that familiarity is selectively impaired by damage to the amygdala[19]. In this study, rats indicated memory for a sample odor by digging in a target cup of sand if it smelled like the remembered sample odor, or by digging in a default cup of sand at the back of the cage if not. Experimenters manipulated the rats' decision criteria by changing the difficulty of digging in the cup and the amount of reward at the bottom, thus allowing them to perform an ROC analysis. In this logic, the biases in difficulty level and reward magnitude in the task for rats function similarly to the confidence ratings as used in studies of ROC curves in humans. Normal rats produced an asymmetrical and curvilinear ROC curve, diagnostic of the combination of recollection and familiarity. In contrast, rats with amygdala lesions produced a flattened but still asymmetric ROC curve, diagnostic of the loss of familiarity. The authors argued that these data suggested that the amygdala was responsible for familiarity, and that the familiarity deficit seen by patient NB[16] was more parsimoniously explained by

her greater amygdala damage than by her lesser perirhinal damage (83% vs 43%, unilaterally).

The connection between the amygdala and memory has received other support. Electrical stimulation of the amygdala during memory encoding improves subsequent retrieval in both rats[20] and humans[21]. In the famous patient SM, who has amygdala calcification due to Urbach-Wiethe disease and blunted fear[22], one of her earliest reported deficits was memory impairment; the authors concluded that "Our case is consistent with the position that the amygdala *is* a crucial component of the neural substrate of memory in humans"[23].

Here, we evaluated this third hypothesis—that the amygdala is necessary for familiarity—using nonhuman primates with selective, fiber-sparing lesions of the amygdala (Fig. 1, Table 1) and four different established tasks that measure familiarity. These include measures of: (1) the increased false alarms to familiar but unstudied lures under a response deadline; (2) the increased errors to highly familiar lures; (3) the modeled estimates of familiarity based on the curvilinearity of ROC curves; and (4) the ability to learn an object discrimination task in which the discriminative cue is the familiarity of objects from previous days. If the amygdala is necessary for familiarity memory, as proposed[19], then monkeys with amygdala lesions should be impaired relative to control monkeys.

## Results

### Experiment 1—Amygdala lesions did not affect familiarity-based error patterns

In Experiment 1, we evaluated whether amygdala lesions affected the time course of familiarity-based errors and the types of errors made under a response deadline. In visual recognition tests, monkeys make a characteristic pattern of errors as a function of response speed[13,24]. The quickest responses show selectively elevated false alarm rates, consistent with selection by a quick but vague familiarity signal. Moderate-speed responses show the lowest rates of both false alarms and misses, consistent with the onset of a slower but more detailed recollective process that can countermand false familiarity signals. Slow responses show the lowest accuracy and elevated rates of both false alarms and misses, characteristic of trials on which the monkeys forgot. Adding a response deadline, which should bias monkeys to use a vague familiarity signal, selectively increases false alarms[24]. Importantly, this pattern in monkeys has been replicated and the errors in the different time periods correlate with more-traditional ROC-based measures of recollection and familiarity in humans[25]. These measures are also very similar to the time course and response deadline measures used with patient NB[16]. Here, we tested whether the natural time course of errors —specifically, the elevated false alarms during the quickest responses— was affected by selective amygdala damage. If the amygdala is necessary for familiarity, then monkeys with selective amygdala lesions

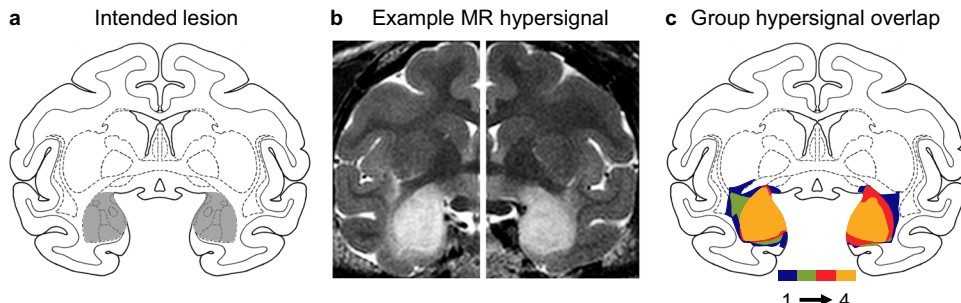

|   | a   Intended lesion | b   Example MR hypersignal | c   Group hypersignal overlap |
|---|---|---|---|

**Fig. 1 | Four rhesus monkeys received bilateral, selective, neurotoxic lesions of the amygdala. a** Diagram of a coronal section of the rhesus monkey brain at the level of the anterior commissure (+17 mm anterior to the auditory canal). The gray shaded region shows the location and extent of the intended lesion. **b** Example images from T2-weighted MR scans from one monkey in the amygdala lesion group acquired ~4 days after surgery in each hemisphere. The white hypersignal over the amygdala indicates edema consequent to injections of neurotoxins. **c** Overlap of observed hypersignal for the four monkeys in the lesion group. Colors indicate signal overlap from 1–4 monkeys.

## Table 1 | Amygdala T2 MRI edema extent

| Monkey | Left | Right | Total |
|---|---|---|---|
| Be | 98.5 | 100.0 | 99.3 |
| En | 98.1 | 92.3 | 95.2 |
| Na | 100.0 | 98.9 | 99.4 |
| Dn | 100.0 | 87.7 | 93.8 |
| MEAN | 99.2 | 94.7 | 96.9 |

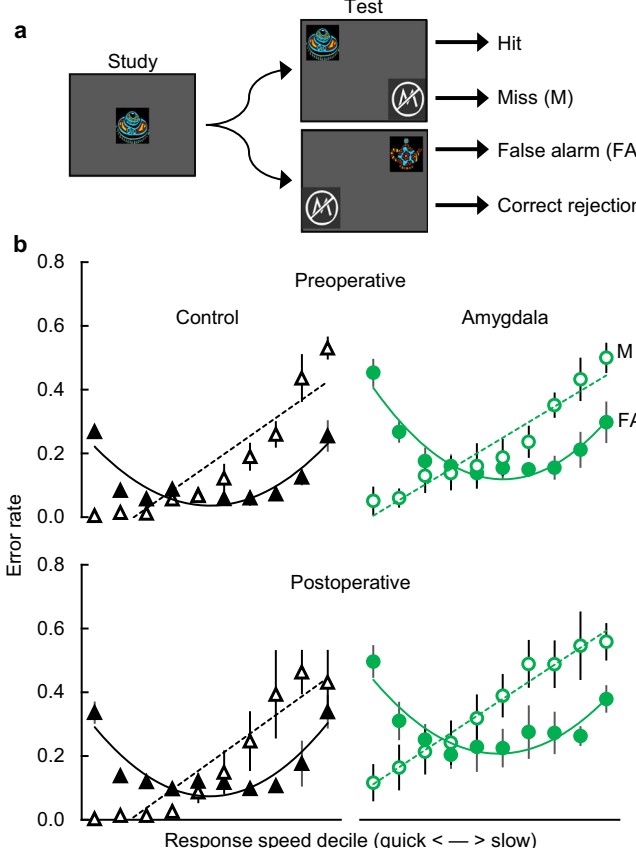

**Fig. 2 | Amygdala damage did not reduce the low-latency false alarms associated with reliance on quick familiarity. a** Diagram of the yes/no recognition test showing a match trial (top) and a nonmatch trial (bottom). Monkeys earned food by touching the test image if it matched the image remembered from study (a hit) or the nonmatch symbol if it did not (a correct rejection). **b** Error rates (±SEM) as a function of response speed decile (10% of responses/bin). Preoperative (top) and postoperative (bottom) performance is shown for control monkeys ($n = 4$; left) and monkeys with amygdala damage ($n = 4$; right). Postoperative data shown here are from the immediate postoperative test with the same stimuli. Closed symbols indicate false alarms (second-order polynomial fit with solid line) and open symbols indicate misses (linear fit with dashed line). Source data are provided as a Source data file.

should show an abnormal time course of errors, likely reflected in fewer false alarms in the quickest responses.

As expected, the pattern of false alarms showed a large effect of response speed (Fig. 2b; $F_{(9, 54)} = 51.78$, $p < 0.001$, partial $\eta^2 = 0.90$, 90%CI[0.83–0.91]), with both groups showing the predicted U-shaped pattern. There was also a main effect of group, with the amygdala group showing slightly more false alarms both before and after surgery ($F_{(1, 6)} = 11.96$, $p = 0.014$, partial $\eta^2 = 0.67$, 90%CI[0.12–0.80]). However, we did not observe an interaction of group with surgical timepoint ($F_{(2, 12)} = 1.04$, $p = 0.384$, partial $\eta^2 = 0.15$, 90%CI[0.00–0.36]) or of group with timepoint and response speed ($F_{(18, 108)} = 0.53$,

$p = 0.014$, partial $\eta^2 = 0.08$, 90%CI[0.00–0.10]), as predicted by the hypothesis that the amygdala should affect familiarity. At the quickest response speed, where we predicted the largest change in false alarm rates, neither group showed any significant change from before to after surgery (amygdala group; $t(3) = -0.87$, $p = 0.447$, $BF_{01} = 2.13$) or rest (control group; $t(3) = -1.00$, $p = 0.390$, $BF_{01} = 1.95$). Instead, we observed robust U-shaped false alarm curves for both groups at both timepoints (Fig. 2b). Thus, amygdala lesions did not produce the disruption in quick familiarity responses predicted by the amygdala hypothesis of familiarity.

### Experiment 2−Amygdala lesions reduced false alarms to highly familiar lures

In Experiment 2, we tested the hypothesis that a disruption in familiarity would manifest as a change in the frequency with which subjects would make false alarms to highly familiar lures. In a yes/no recognition task, normal monkeys will make some baseline level of false alarms to unstudied lures, and this false alarm rate will significantly increase on probe trials for which the unstudied lure was seen more recently than the typical lure[13]. This increase in false alarms to highly familiar lures is greater after longer retention intervals than after shorter retention intervals, consistent with memory being more fragile and susceptible to false familiarity after longer forgetting periods. If the amygdala is necessary for familiarity, then monkeys with selective amygdala damage should show an abnormal rate of false alarm to highly familiar lures.

As expected, monkeys made significantly more false alarms on probe trials with the highly familiar lure than on baseline trials with the normally-familiar lure (Fig. 3b; main effect of trial type: $F_{(1, 6)} = 63.80$, $p < 0.001$, partial $\eta^2 = 0.91$, 95%CI[0.62–0.96]) and significantly more false alarms on the long delay trials than on the short delay trials (Fig. 3b; main effect of delay length: $F_{(1, 6)} = 31.66$, $p = 0.001$, partial $\eta^2 = 0.84$, 95%CI[0.38–0.93]). Consistent with the hypothesis that amygdala damage should affect false alarms to highly familiar lures, the effect of trial type depended both on surgical timepoint and on surgical group (interaction between trial type, timepoint, and group: $F_{(1, 6)} = 14.23$, $p = 0.009$, partial $\eta^2 = 0.70$, 95%CI[0.10–0.88]). Full ANOVA results are in Table S1. In our planned critical comparisons, this manifested as lower false alarms to probe trials with the longer delay length for the amygdala group postoperatively relative to preoperatively ($t(3) = 5.19$, $p = 0.014$, $d = 2.59$, 95%CI[0.46–5.51], $BF_{01} = 0.15$). This change in false alarm rates on probe trials preoperatively to postoperatively was not seen for the control group in any condition and did not reach statistical significance with the amygdala group for the shorter delay (all $p > 0.31$, all $BF_{01} = 1.65–2.79$). As noted above, these findings are consistent with a role for the amygdala in familiarity; when confronted with highly familiar lures, the amygdala group appeared to experience less false familiarity. However, this interpretation of impaired familiarity in Experiment 2 conflicts with the interpretation of spared familiarity in Experiment 1. In Experiments 3 and 4, we employed two additional tests of familiarity to provide additional evidence.

### Experiment 3−Amygdala lesions did not affect ROC curves
Experiment 1 showed a normal time course of false alarms, suggesting spared familiarity, but Experiment 2 showed reduced false alarms to abnormally familiar lures, suggesting impaired familiarity. As one of two tests to adjudicate between these competing interpretations, we compared the familiarity and recollection estimates derived from ROC curves. In normal tests of visual recognition, monkeys produce asymmetrical and curvilinear ROC curves[26], similar to those in humans[27] and rats[7]. In all three species, the asymmetry has been interpreted as measuring the contribution of recollection and the curvilinearity has been interpreted as measuring the contribution of familiarity (Fig. 4b)[28], but see ref. 29. Indeed, a flattening of the ROC curve was shown by Patient NB, who has a deficit in familiarity[16], and

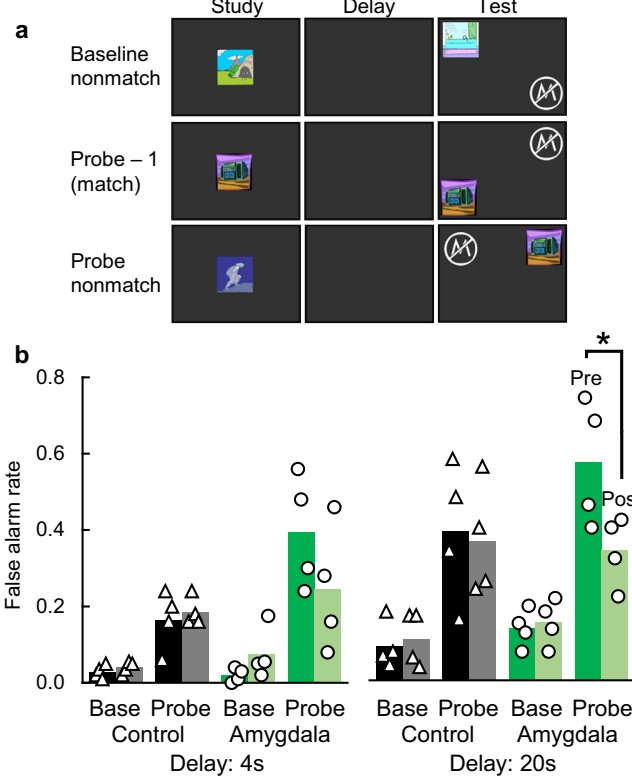

**Fig. 3 | Amygdala damage reduced false alarms to highly familiar probe lures.** **a** Example screens from the yes/no recognition task. Baseline match and nonmatch trials proceeded as in Exp 1. Probe trials were the same as nonmatch trials except that the to-be-rejected lure was the sample from the previous trial. **b** False alarm rates as a function of surgical timepoint (preoperative or postoperative on left and right, respectively, of each pair of bars), trial type (baseline or probe), group (control, $n = 4$, or amygdala, $n = 4$), and retention interval (4 or 20 s). Bars show group means and points show individual monkeys. * indicates $p = 0.014$ via two-sided, uncorrected, paired t-test. Source data are provided as a Source data file.

was taken as evidence of a loss of familiarity in the rats with selective amygdala lesions[19]. Thus, we tested monkeys on a variant of Guderian's ROC recognition test[26]. If the amygdala is necessary for familiarity, then monkeys with selective amygdala lesions should show flattened ROC curves like those of Patient NB and of rats with amygdala lesions.

Groups did not differ in overall accuracy (mean d': control = 2.70, amygdala = 2.58; $t(6) = 0.32$, $p = 0.759$). Decision criteria differed significantly over the bias levels, as intended (Fig. 4c; $F(4, 24) = 60.67$, $p < 0.001$, partial $\eta^2 = 0.91$, 90%CI[0.84−0.94]) but not between groups (Fig. 4c; main effect of group: $F(1, 6) = 0.44$, $p = 0.532$; interaction: $F(4, 24) = 0.67$, $p = 0.620$). Critically, groups did not differ in parameter estimates of either recollection or familiarity (Fig. 4d; Table 2; recollection: $t(6) = 0.59$, $p = 0.579$, $BF_{01} = 2.06$; familiarity: $t(6) = 0.62$, $p = 0.556$, $BF_{01} = 2.03$). For comparison, the means and ranges of the recollection and familiarity parameters from intact monkeys in the previous study[26] are also presented in Table 2. The spared familiarity estimates of monkeys with selective amygdala lesions stand in stark contrast with the impaired familiarity estimates of the rats with selective amygdala lesions[19] and of Patient NB[16]. Thus, the evidence from ROC curves bolsters the evidence from Experiment 1 that amygdala damage does not impair familiarity.

### Experiment 4—Amygdala lesions did not affect across-day familiarity
Experiments 1–3 provided one piece of evidence that amygdala lesions impaired familiarity and two pieces of evidence that they left

familiarity intact. As a final piece of evidence to adjudicate between these two interpretations, we tested monkeys on Browning's Constant Negative test of familiarity discrimination[30]. This task is an S+/S− discrimination task in which the discriminative cue is the repetition of S− stimuli across testing days. It has been interpreted as a familiarity discrimination and is left spared by the type of prefrontal-temporal disconnection that impairs more complex memory for lists of items[30] but is impaired by neonatal perirhinal cortex lesions[31]. Interestingly, Patient NB also shows an impairment in cumulative lifetime familiarity[32] and the build-up of item experience across days in the Constant Negative task might test familiarity in a similar way. If the amygdala is necessary for familiarity, then monkeys with selective amygdala lesions should show blunted learning of this across-day familiarity discrimination.

In the critical comparisons, the two groups did not differ in sessions to criterion for any of the three problem sets (Fig. 5B; set 1: $t(6) = 1.03$, $p = 0.343$, $BF_{01} = 1.63$; set 2: $t(6) = -1.51$, $p = 0.182$, $BF_{01} = 1.14$; set 3: $t(6) = -2.10$, $p = 0.080$, $BF_{01} = 0.69$). The learning slope did become steeper across the problem sets, as shown by a significant main effect of set ($F(2, 12) = 6.708$, $p = 0.011$, partial $\eta^2 = 0.528$, 90% CI[0.10−0.67]); however, the slope of the learning did not by differ between groups ($F(1, 6) = 1.15$, $p = 0.324$, partial $\eta^2 = 0.161$, 90% CI[0.00−0.49]) and there was no group × set interaction ($F(2, 12) = 1.92$, $p = 0.189$, partial $\eta^2 = 0.243$, 90%CI[0.00−0.45]). Thus, we found no evidence that amygdala lesions impaired acquisition of an across-day familiarity discrimination.

### Experiment 5—Amygdala lesions impaired reward processing
In Experiments 1–4, three out of four pieces of evidence pointed to spared familiarity following amygdala damage. However, one piece of evidence suggested impaired familiarity: reduced false alarms to highly familiar lures. To understand this anomaly, we considered non-mnemonic factors. As indicated earlier, our leading mnemonic explanation was that monkeys with amygdala lesions made fewer false alarms to recently seen lures because they experienced less increased familiarity than did normal monkeys. However, another possible explanation considers the reward history of the lure, in addition to its familiarity. According to this account, normal monkeys made false alarms to recently seen lures because those lures were both recently seen and also recently rewarded. On the probe nonmatch trials, the lure had been seen on the previous match trial, had likely been rewarded, and had thus acquired positive associative strength. On the probe trial, the choice of that lure may have been a combination of the false familiarity from having recently seen the lure and the associative strength from having recently been rewarded for selecting the lure. If so, a decrement in choosing it on the probe trial could be a result of an impairment in familiarity or an impairment in reward processing, or both. Such an impairment in reward processing might be due to an impairment in the speed of reward learning, the consistency of reward-guided choices, or both. To test this alternative hypothesis about impaired reward processing, we tested monkeys in an established paradigm that requires rapid reward association learning: a multi-arm bandit explore-exploit decision making task[33]. Similar to one-armed bandit gambling machines, this task is a three-armed bandit, in which three images are paired with three probabilities of reward (Fig. 6a). Critically, one of the images is replaced regularly, the new image is assigned a reward probability, and the monkey must quickly learn through trial and error whether it is better, worse, or similar to the remaining images. A wealth of evidence ties the amygdala to various aspects of reward processing[34,35] and monkey amygdala neurons encode multiple parameters of this particular reward-processing task[33]. If the reduced false alarms in Experiment 2 were due to impaired reward association processing, then monkeys with amygdala lesions should show abnormal reward processing in this explore/exploit task.

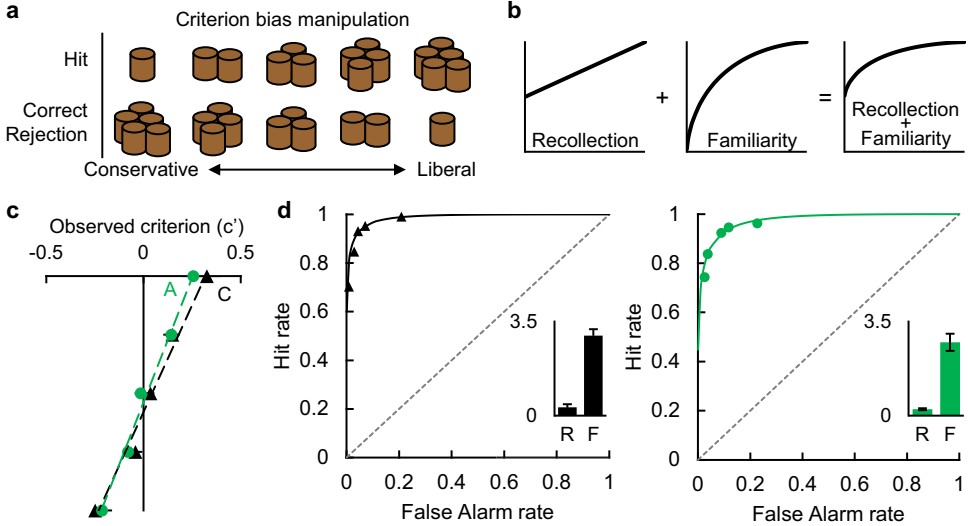

**Fig. 4 | Amygdala damage did not change ROC measures of recollection or familiarity. a** We manipulated the decision criterion of the monkeys by changing the amount of food reward for hits and correct rejections each day. For example, five pellets for a correct rejection and one pellet for a hit should produce a very conservative bias to respond 'no' when uncertain. **b** The dual-process signal detection model posits that detailed recollection produces a flat but asymmetrical ROC curve, vague familiarity produces a curved but symmetrical ROC curve, and normal recognition produces a curved and asymmetrical ROC curve that represents the combination of recollection and familiarity. **c** The mean observed criterion bias (c' ±SEM) of our monkeys as a result of the food manipulations in panel **a**. Negative numbers are more conservative biases and positive numbers are more liberal biases. These data indicate that the decision criteria of our monkeys were affected by the reward manipulations, as intended. Monkeys with amygdala damage ($n = 4$) in green circles and control monkeys in black triangles ($n = 4$). **d** The ROC curves for both control (left, black triangles, $n = 4$) and amygdala (right, green circles, $n = 4$) monkeys were curved and asymmetrical. Points represent group means of hit and false alarm rates at the five criterion bias levels. Inset bar graphs depict the mean (±SEM) of the individual monkeys' parameter estimates for recollection (R) and familiarity (F). Source data are provided as a Source data file.

The monkeys with amygdala lesions did not modulate their exploration of novel choice options until trial 14, but only when we did not correct for multiple comparisons. When we applied a stricter criterion, the amygdala lesion group did not show evidence of reliable reward-dependent exploration (Fig. 6c). This deficit seemed to be driven primarily by an inability to discriminate between stimuli that predicted the medium and high reward probabilities. In contrast, the control group's choices diverged to reflect the assigned reward probability within the first 7 trials, corrected for multiple comparisons (Fig. 6c; all $p < = 0.015$), after which the control monkeys' exploration of the novel options was reliably reward dependent. Notably, both the control and amygdala lesion groups showed an initial avoidance of novel choice options (Fig. 6c, leftmost points) and appeared to explore them at a similar rate. This implies that amygdala damage did not impair the monkeys' ability to discriminate novel from familiar options or their capability to learn and update the value of novel and familiar choice options. Rather, the amygdala lesion group appeared to have a deficit in using learned values to consistently select the most rewarding option.

These hypotheses were confirmed when we examined group differences in the fitted RL model parameters. The novelty bonus was negative and did not differ between the lesion and control groups (Fig. 6b; $F(1, 6) = 0.03$, $p = 0.874$, partial $\eta^2 = 0.005$, 90%CI[0.00–0.20], $BF_{01} = 1.47$), suggesting a normal ability to discriminate familiar images from novel images. The learning rates were similar in the control and amygdala lesion groups (Fig. 6b; $F(1, 7) = 0.04$, $p = 0.848$, partial $\eta^2 = 0.006$, 90%CI[0.00–0.20], $BF_{01} = 2.21$); however, we did find that, compared to the control group, the inverse temperature parameter was significantly reduced in the amygdala lesion group (Fig. 6b; $F(1, 6) = 7.70$, $p = 0.035$, partial $\eta^2 = 0.581$, 90%CI[0.03–0.75], $BF_{01} = 0.72$). This suggests that amygdala damage produced an impairment in reward processing that led to inconsistent selection of the most valuable choice option. We have previously found the same deficit in a separate group of monkeys with amygdala lesions in a reversal learning task[36].

## Discussion

Across four paradigms, the bulk of the evidence converges on the conclusion that the amygdala is not necessary for the familiarity aspect of recognition memory. Monkeys with selective amygdala lesions showed a normal time course of quick false alarms, normal curvilinear and asymmetric ROC curves with normal parameter estimates of familiarity and recollection, and normal learning of an across-day familiarity discrimination. The one piece of evidence that might have pointed to a familiarity deficit, decreased false alarms to recently seen lures, can be accounted for by a deficit in reward processing. In support of this interpretation, we observed a deficit in reward processing in the amygdala lesion group, validating this explanation and the effectiveness of the lesions. Indeed, the fact that amygdala damage did

## Table 2 | ROC parameter estimates for recollection and familiarity

|  | Monkey | R | F |
|---|---|---|---|
| Amyg | Be | 0.146 | 2.501 |
|  | En | 0.282 | 2.881 |
|  | Na | 0.322 | 3.525 |
|  | Dn | 0.222 | 2.005 |
|  | MEAN | 0.243 | 2.728 |
| Con | Al | 0.434 | 2.921 |
|  | Ds | 0.290 | 3.491 |
|  | Ta | 0.002 | 3.143 |
|  | Ar | 0.530 | 2.353 |
|  | MEAN | 0.314 | 2.977 |
| Guderian et al.[26] | Mean | 0.398 | 1.504 |
|  | Range | 0.062–0.757 | 1.041–2.594 |

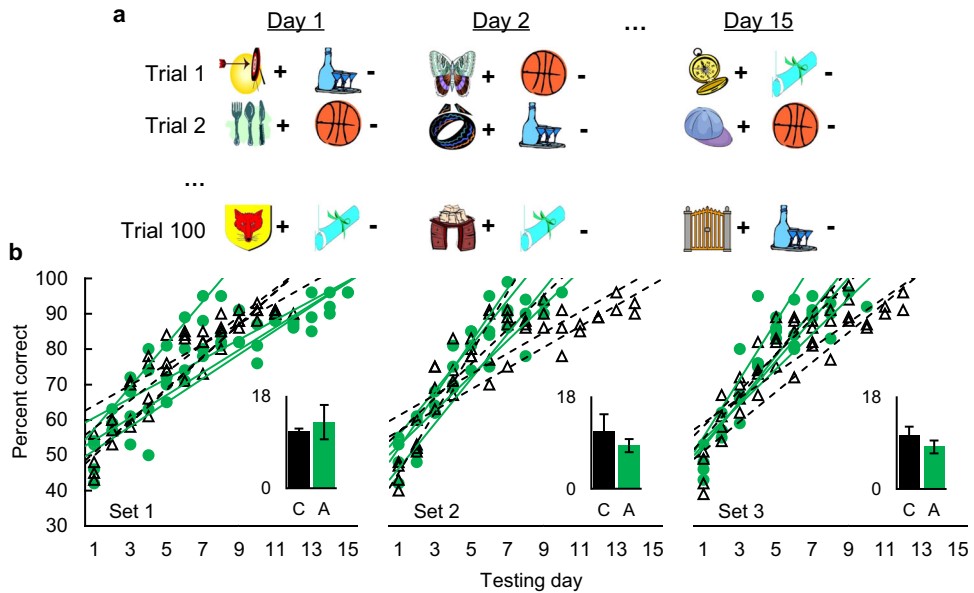

**Fig. 5 | Amygdala damage did not impair across-day familiarity discrimination.**
**a** Example stimuli used in each two-choice trial; + denotes a rewarded image and –
denotes a nonrewarded image. The familiarity discrimination was such that novel
images were always rewarded and familiar images were never rewarded (except for
on Day 1, when all images were novel and reward assignments were initially

unknown to the monkey). **b** Percent correct as a function of group (black triangles
with dashed lines = control, $n = 4$; green circles with solid lines = amygdala, $n = 4$),
testing day, and problem set. Each line is a linear fit for an individual monkey. Inset
bar graphs depict the mean sessions to criterion (±SD) for each group (C = control,
A = amygdala) and problem set. Source data are provided as a Source data file.

not affect the valuation of novelty in the reward learning task further
supports the position that the amygdala is not critical for detecting
item familiarity/novelty or discriminating between familiar and
novel items.

These data constrain the possible hypotheses about the neural
bases of familiarity. As outlined in the Introduction, the familiarity
aspect of recognition has been variously attributed to the perirhinal
cortex[16], the entorhinal cortex[18], and the amygdala[19]. This amygdala
hypothesis of familiarity has gained some traction in subsequent
review papers[37–40]. However, memory researchers express widely ran-
ging views of amygdala contributions to familiarity. Some investiga-
tors claim a strong role for the amygdala in familiarity[19], some suggest
the amygdala is one of multiple structures whose damage can impair
familiarity[39], some posit that the impairment following amygdala
damage is due to the loss of amygdala input to the perirhinal cortex[37],
and others simply say the role of the amygdala is controversial[40]. Our
data fully contradict the amygdala hypothesis of familiarity. Instead,
they suggest that models of recognition memory do not need to
incorporate the amygdala as a structure necessary for familiarity.

Our data help interpret prior findings from patients with MTL
damage. In the first patient to show a selective familiarity deficit, NB,
her most extensive damage was to the amygdala and yet the authors
attributed her impairments to her perirhinal cortex damage[16,17]. This
attribution was based in part on the lack of familiarity deficits in other
patients whose damage included the amygdala but not the perirhinal
cortex[17]. In our study, the relative sparing of the perirhinal cortex
(estimated damage = 3.0%) and lack of robust familiarity deficit is
consistent with the hypothesis that Patient NB's familiarity deficit
might be due to her perirhinal damage[16,17]. More concretely, our study
strengthens the case that Patient NB's familiarity deficit should not be
attributed to her substantial amygdala damage.

Our study is also consistent with the broader literature on how to
characterize the deficits seen in human patients with amygdala
damage. Although there are some reports of general memory issues
with neutral stimuli in some patients with amygdala damage[23,41], there
are also reports in which these patients demonstrate normal memory
for neutral stimuli[41,42]. Instead, the most robust deficits seem to be in

non-mnemonic tasks such as in processing emotions, or in mnemonic
tasks that involve emotional stimuli[22,42,43]. The current data add to the
evidence that amygdala damage is generally not accompanied by
general memory impairments.

It is unclear how to reconcile our finding that the monkey amyg-
dala is not necessary for familiarity across multiple paradigms with the
previous finding that the rodent amygdala is necessary for familiarity
in an ROC paradigm[19]. One possibility is that the discrepancy is due to
the different anatomy of the primate and rodent amygdala. The
amygdala is conserved across species in many ways[44,45]. However, the
monkey amygdala is not a uniformly scaled-up rodent amygdala: the
central and medial nuclei show a 4–8× expansion, whereas the
lateral, basal, and accessory basal nuclei show a 32–39× expansion
relative to rodents[46]. Some of this expansion is due to the increased
number of connections between the basolateral region of the amyg-
dala and prefrontal cortex that accompanied emergence of new pre-
frontal cortex regions in the primate lineage[47]. In rats, the paralaminar
nucleus is situated laterally and is small enough that many rat brain
atlases do not mention it, but in monkeys and humans it is situated
ventrally and is extensive[48]. Other anatomical differences exist, such as
in the extent of convergence of cortical projections[45] and primate-
unique profiles of glutamatergic, GABAergic, and astroglia subtype
cells[49]. A second possibility is that the discrepancy is due to the
methodological differences between how rodent and primate memory
are tested. The rodents remembered odors, had to physically move to
different locations to make accept or reject responses, and had their
response criterion manipulated through a combination of response
difficulty and reward magnitude[19]. In contrast, our monkeys had to
remember visual images, made responses from a single seated posi-
tion, and had their response criterion manipulated through reward
magnitude alone. Research demonstrates that manipulations of effort
and reward are not interchangeable in their effect on behavior or
equivalent in how much they involve the amygdala[50]. Further, although
basic mechanisms of reinforcement learning often seem universal,
they actually often differ across species[51,52]. A third possibility is that
the rodent results[19] represent a false positive. However, in the absence
of contradictory evidence from a preregistered replication attempt or

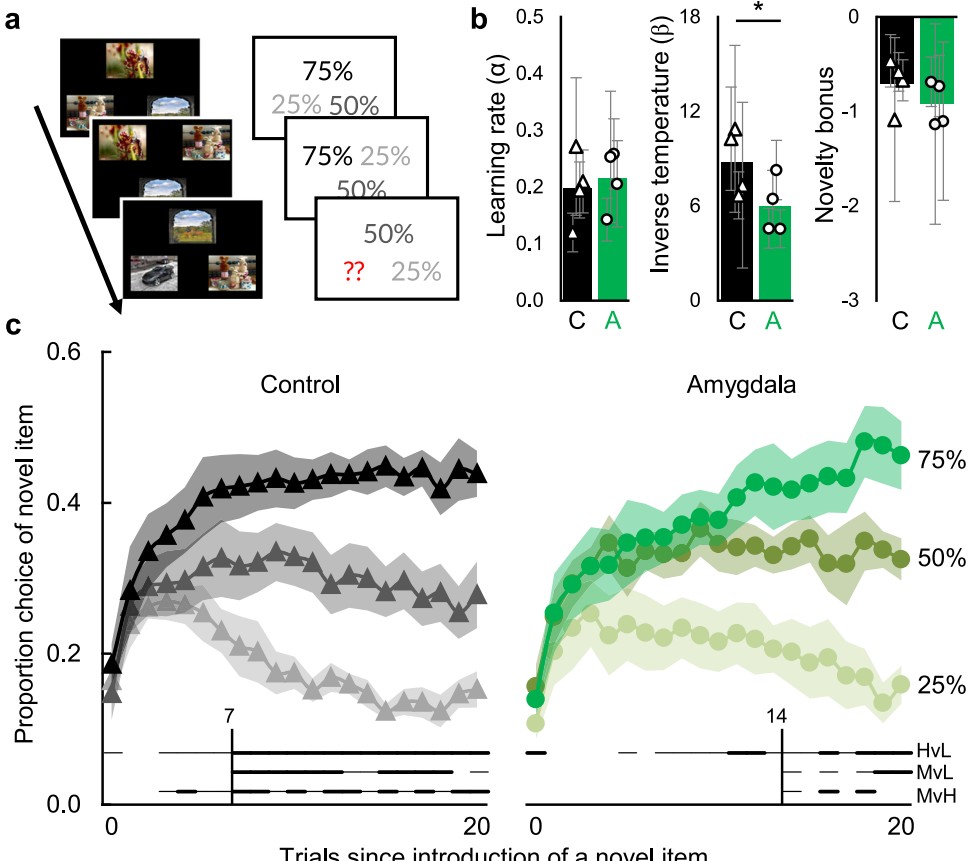

**Fig. 6 | Amygdala damage impaired rapid reward association learning. a** Trial schematic. Examples screens (left) show the stimuli as they appeared to the monkey. Probabilities (right) show the pre-assigned reward probabilities of the images. The red question marks indicate a novel stimulus with an unknown (to the monkey) probability that must be learned. **b** Model-derived learning parameters for control (black bars, *n* = 4) and amygdala (green bars, *n* = 4) monkeys. Learning rate (α) quantifies how much learning occurs from each outcome; Inverse temperature (β) quantifies the consistency with which subjects select the most valuable option; Novelty bonus quantifies the bias to choose or avoid a novel item with an unknown reward probability. Bars are group means and points are parameters (±95% CI) for individual monkeys. * indicates *p* = 0.035 via linear mixed-effects model as

described in text. **c** Proportion (±SEM) of times a novel option was chosen as a function of assigned reward probability (75%, 50%, or 25%) and number of trials since the introduction of a novel item. Trial 0 is the trial on which the novel item was introduced. Horizontal lines below the data indicate trials for which two conditions were chosen at statistically different rates (uncorrected = thin line; Bonferroni corrected = thick line; HvL = high reward vs. low reward; MvL = medium reward vs. low reward; MvH = medium reward vs. high reward). Vertical lines indicate the trial on which all three conditions were chosen at statistically different levels (uncorrected). Compare panel **c** to Fig. 1E from Costa et al. (2019). Source data are provided as a Source data file.

multiple converging paradigms, we take their results at face value. Thus, determining whether some of these species and/or methodological differences might have contributed to the different results will be a challenge.

These results bolster the evidence linking the amygdala to reward processing. Using a three-arm bandit explore-exploit task, we found that the amygdala was critical for maintaining consistent choice of the high value items. Broadly, this is consistent with the wider literature examining amygdala contributions to visual learning[34,35,53]. Specifically, it is most similar to prior work using a two-arm bandit reversal learning task, which also found that amygdala damage impaired choice consistency[36]. The observation that amygdala damage impairs value-based choice consistency across two separate reward processing tasks with different task demands should help guide future research.

We also observed two inconsistencies with prior work on reward processing, both of which likely stem from differences in task demands or modeling approaches. First, we found that monkeys were initially novelty averse. In contrast, prior studies that used an oculomotor version of this same paradigm found that monkeys were initially novelty prone (compare Fig. 6c to Fig. 4B from ref. 54 and Fig. 1E from ref. 33). This apparent discrepancy might be due to the different

response requirements: our monkeys selected an option by touching the image, whereas monkeys in prior studies made their choice by looking at the image. Thus, in the oculomotor task, the novelty biases in visual exploration and in choice behavior are confounded—a monkey cannot look at the novel item without selecting it. It is worth noting that a touchscreen variant of the same oculomotor task did find that humans were novelty prone[55]. But unlike in the current experiment, the task was not self-paced and participants were asked to make confidence judgments prior to receiving feedback. If the response demands of the task (i.e., oculomotor vs touchscreen) change the monkeys' early experience with the value of novel stimuli, they may also change the estimates of their learning parameters. Future work modeling choice behavior and novelty preference might benefit from separating the effects of novelty biases in visual exploration and in manual choice. Second, we found no effect of amygdala lesions on the learning rate parameter when tested on the three-arm bandit explore-exploit task, whereas prior work did find amygdala lesions impaired learning from positive feedback in a two-arm bandit reversal learning task[36]. This is likely explained by task differences and the way the models capture behavior. The model used to fit behavior in the reversal learning task specified two separate learning rate parameters, one for learning from gains and one from losses, because there were

only two reward contingencies and they were anticorrelated[36], whereas this was not theoretically justified with the explore-exploit task used here because there were three options and choice of one option no longer gave the monkeys perfect information about the value of the other options[33,54]. Further, the learning rate parameter in the model is most influenced by the first few choices of the novel item and thus it will produce a better-fit estimate of early learning rates with novelty-prone subjects than with novelty-averse subjects because they contribute more trials to the model. However, the model is equally good in the two tasks at capturing the inverse temperature parameter during asymptotic performance, and this measure of choice consistency is where we observed the same deficit as found in the reversal learning task[36].

One potential explanation for why a deficit in reward processing affected Experiments 2 and 5, but not Experiments 1, 3, and 4, might lie in the amygdala's proposed role in noncontingent statistical learning[56,57]. When learning about rewards, organisms learn not only the causal contingencies of which action/choice produced a reward, but also statistical regularities about noncontingent actions/choices/cues that occurred in the penumbra of the reward. For example, when you select a specific image and are rewarded, you learn that choice/reward contingency, but that association might spread to the nearby, noncontingent things such as your choice on the previous trial. When an organism completes the same action repeatedly, contingent and noncontingent learning are usually consistent. But when contingencies shift, when associations are still fragile during initial learning, or when one must learn about multiple probabilistic outcomes, contingent and noncontingent learning can be inconsistent. In humans and monkeys, activity in the amygdala has been found to track this type of noncontingent learning[56,57]. In Experiment 5, monkeys needed to rapidly learn the statistical regularities of multiple probabilistically rewarded choices while associations were still fragile and uncertain, so the influence of noncontingent learning on behavior was likely high. Removing this noncontingent learning about statistical regularities might have resulted in irregular choices, manifesting in the observed lower choice consistency. In Experiment 2, for which images were also relatively novel and so associations and visual memory traces were still fragile, the to-be-rejected probe lure was irrelevant to the current trial and actually inconsistent with the memory trace from the current trial. Removing this inconsistent reward association would improve performance, manifesting in the observed lower error rate after surgery. In Experiment 1, all targets and lures were highly familiar and had likely acquired similar histories of noncontingent learning, limiting the influence of noncontingent learning on behavior. For Experiment 3, all items were initially novel and items that had acquired both contingent and noncontingent learning in previous trials were never brought back as lures, limiting the influence of noncontingent learning on behavior. For Experiment 4, only the unrewarded items were brought back for subsequent sessions, limiting the influence of noncontingent learning on behavior. In short, the experiments in which we see an effect of amygdala lesions are those in which we might predict the largest role of noncontingent learning in intact animals. Consistent with this idea, a large analysis of monkeys with a variety of focal lesions found that lesioning the amygdala and areas interconnected to it is necessary for using prior beliefs about environmental reward statistics to stabilize value representations when learning in dynamic environments[58]. Thus, this explanation based on noncontingent learning merits further study.

Methodologically, investigating the hypothesized role of the amygdala in familiarity through the use of multiple established paradigms provides converging evidence that allows us to draw particularly strong conclusions. Indeed, the converging evidence from multiple paradigms is one of the strengths of the initial report of a familiarity deficit[16,32]. Regarding the current study, the time course of early false alarms has been used multiple times with monkeys[13,24] and,

critically, has been replicated in monkeys by independent researchers and shown to correlate with the more established method of acquiring ROC measures of familiarity in humans[25]. ROC curves have been well established as providing consistent evidence between humans and nonhumans[28] and have provided good evidence of the dual-processes of recollection and familiarity in monkeys[26]. The Constant Negative test of familiarity discrimination has also been used with monkeys from different independent laboratories[30,31] and been shown to be sensitive to neonatal perirhinal cortex damage[31], which is consistent with the perirhinal hypothesis of familiarity. Each of these tasks has its respective weakness, but those weaknesses are not overlapping. Thus, the converging evidence from these tasks should promote confidence in our conclusions.

The biggest limitation of the current study is that we cannot gather subjective measures of familiarity. In humans, one of the simplest ways to measure if somebody recollects an item or merely finds it vaguely familiar is to ask them if they remember it in detail or just know that it was seen previously[59]. Such remember/know paradigms have been enormously influential in studies of recollection and familiarity[1,60] and provide much of the evidence for the original report of a familiarity deficit[16]. However, monkeys do not speak, and therefore cannot tell us about the subjective feel of their memories. We must infer that they are experiencing a vague sense of familiarity from their tendency to falsely accept lures under circumstances where those lures should be rejected but might be accepted if remembered through vague familiarity. This weakness is partially circumvented by using multiple converging paradigms, but it will always remain a weakness in studies of nonhuman subjects. One additional limitation of this study is that our Bayes Factor analyses consistently revealed $BF_{01}$ values under three for null results, which is relatively low support for the null hypothesis compared to the standards of the field. This is likely due to the sample size often used with studies of nonhuman primates. Thus, although the power of our Bayesian analysis is a limitation of this study, it is consistent with the frequentist analysis, consistent across most paradigms, and directly counter to the amygdala hypothesis of familiarity.

In conclusion, these results contradict prior work proposing a critical role for the amygdala in the familiarity aspect of recognition memory[19]. They also provide guidance in interpreting the case studies of familiarity impairments following brain damage[16,32]. Across multiple established paradigms, the preponderance of the evidence suggested that monkeys with selective amygdala lesions had normal familiarity. Instead, we found a predicted role of the amygdala in reward processing, specifically in maintaining consistent choices of high value items. Accordingly, we conclude that the amygdala is not necessary for the familiarity aspect of recognition memory.

## Methods

### Subject & apparatus

Eight adult male rhesus monkeys (*Macaca mulatta*; mean age at start of testing 4.93 years) served as subjects. All eight monkeys were naive to cognitive testing. The monkeys were housed with protected social contact, which allowed physical grooming through a perforated barrier with one familiar partner, and had visual and auditory contact with several other conspecifics. They were kept on a 12-h light-dark cycle and had ad libitum access to water. Food was controlled to maintain motivation with monkeys' weights remaining above 85% of their free-feeding weight. Monkeys were tested five days a week for 2-h sessions in sound-attenuated testing booths (BRS/LVE, Laurel Maryland) that contained a 15" touchscreen, generic audio speakers, and two food-pellet dispensers (Med Associates, St. Albans, VT). Correct responses were rewarded with a random mix of nutritionally complete food pellets and flavored sucrose pellets in a 95:5 ratio. This study was carried out in accordance with the Guide for the Care and Use of Laboratory Animals and the US Animal Welfare Act. All procedures

were reviewed and approved by the National Institute of Mental Health Animal Care and Use Committee.

## Surgery

Half of the monkeys ($n = 4$) were pseudo-randomly assigned to receive selective, excitotoxic bilateral amygdala lesions (mean age at surgery 5.16 years). Surgical procedures for producing selective lesions of the amygdala via MR-guided injection of excitotoxins have been previously described[15], e.g., refs. [61–63]. Briefly, stereotaxic coordinates for a series of injection sites were calculated for each monkey based on their T1-weighted MRI scan. Injection sites were tailored to each subject's amygdala to both maximize coverage of the amygdala and minimize damage to neighboring structures (16–19 injection sites/hemisphere; 0.6–1.2 µl/site of ibotenate). Monkeys were anesthetized with ketamine (10 mg/kg; intramuscular injection) and then maintained with isoflurane gas (1–3% to effect). Throughout surgery, we monitored vitals including blood pressure, respiratory rate, heart rate, temperature, blood oxygen saturation, and exhaled/inhaled $CO_2$. For each injection site, a 30-gauge Hamilton syringe needle was lowered to the desired coordinates and we expressed 0.6–1.2 µl of ibotenate (10 mg/ml) at 0.2 µl/min. To reduce the risk of postoperative complications due to edema, bilateral lesions were carried out in two surgeries that were separated by a minimum of two weeks. All surgeries were carried out under aseptic conditions.

Control monkeys were unoperated. Each control monkey was randomly matched with one of the operated monkeys and rested a number of days equal to the total time between that monkey's last preoperative testing day and first postoperative testing day (mean = 51 days). During this interval, there was no cognitive testing; food and water were available ad libitum.

## Lesion assessment

Lesions were confirmed using a T2-weighted MRI scan acquired approximately 4 days after surgery[61,64]. The extent of the white hypersignal diagnostic of edema and cell death was traced manually onto a standard template brain at 1 mm increments and then the rough extent of the affected area was calculated by summing the volume of the amygdala covered by hypersignal across the depth of the amygdala. The hypersignal generally covered the majority of the amygdala (mean = 96.9%), with relatively little involvement of surrounding structures, and consistency between hemispheres and monkeys was high (Fig. 1; Table 1). Within the amygdala, hypersignal coverage and lesion extent are not related in a 1:1 ratio[61]. Based on data comparing the extent of hypersignal in T2 MRIs following amygdala lesions with the extent of damage observed in post-mortem histological examination[61], we can conclude two things: (a) the nearly full hypersignal coverage of the amygdala indicates that the true damage is greater than 50% bilaterally and (b) based on the generally high damage seen with these techniques (82.4%), the true damage in this study is likely much higher than 50%. Total unintended damage was generally low across surrounding structures and usually unilateral. Based on regression functions described previously[61], unintended damage was estimated as follows: hippocampus: hypersignal coverage = 9.7%, estimated damage = 4.0% (95%PI: 0.0–11.9%); entorhinal cortex: hypersignal coverage = 15.0%, estimated damage = 5.7% (95%PI: 0.0–20.0%); perirhinal cortex: hypersignal coverage = 8.9%, estimated damage = 3.0% (95%PI: 0.1–7.8%). Thus, the combination of substantial bilateral damage to the amygdala and minor, unilateral damage to surrounding structures should produce a valid test of the necessity of the amygdala for these tasks.

## Statistics and reproducibility

Individual statistical tests are described for each experiment. Sample size was chosen to be consistent with the conventions of the field and other studies of lesions in nonhuman primates. Sex was not included as a variable because the NIH Policy on Sex as a Biological Variable states that nonhuman primates represent an acutely scarce resource which justifies limiting the number of animals used. Data exclusions are described for each experiment. They are: For Experiment 3, 7% of data were excluded because they did not pass the manipulation check. For Experiment 5, 3.4% of data were excluded for the modeling analysis, but not the other analyses, because they were from sessions in which the modeled parameters did not converge. Subjects were randomly assigned to groups. Data were collected by computers and the experimenters were not present during testing, eliminating the potential for unblinded experimenters to influence subject's performance during testing sessions.

## Experiment 1

**Training procedure.** Prior to data collection and surgery, monkeys were trained to use a touchscreen via standard autoshaping procedures and then to complete a yes/no recognition task as depicted in Fig. 2a and described previously[24]. Each correct response (hit or correct rejection) was rewarded with one food/sugar pellet and each mistake (miss or false alarm) was followed by a 2-s time out. As monkeys reached >75% correct criterion at each stage, we increased the delay between study images and choice test (0–4 s, increasing at 0.5 s increments) and then subsequently decreased the image set size (800, 100, 40, 20, 3, and 2 images). Intertrial intervals (ITI) were 10 s. To prevent accidental selection, responses for pre-training and all subsequent experiments required two consecutive touches in the same location on the screen. Training criteria were reached when monkeys completed three consecutive sessions of 400 trials at a 4-s delay with an image set size of two with at least 75% accuracy. Data from this training set was not used in experimental data analysis.

**Task procedure.** Prior to surgery, monkeys completed two 1000-trial sessions of a yes/no recognition task (Fig. 2a). Target stimuli were two color clipart images (300 × 300 pixels) chosen to be visually similar and thus to elicit a high amount of baseline familiarity. We used a small set of stimuli because smaller image sets are more difficult to remember, thus producing enough errors for an error analysis, and because repeating images should enhance the baseline familiarity of each image, thus requiring monkeys to use detailed recollection to distinguish between targets and lures and producing elevated levels of familiarity-based errors[24]. Monkeys initiated a trial by touching a green start box at the bottom of the screen (100 × 100 pixels), saw one sample image, and then touched it to progress. Sample stimuli were chosen pseudo-randomly such that each image was the sample equally often within each block of eight trials. After a 4-s retention interval, monkeys were presented with one test image and a nonmatch symbol. These images appeared pseudo-randomly in two of four locations on the screen each trial to prevent monkeys from developing a location bias. If the test image was the same as the previously seen sample image, monkeys could earn a food reward by touching that test image (match trial). If the test image was different than the previously seen sample image, monkeys could earn a food reward by touching the nonmatch image (nonmatch trial). To prevent monkeys from developing a bias, match, and nonmatch trials were seen equally often. Correct trials always produced a secondary audio reinforcer ("excellent!" or "woo-hoo!"). The ITI was 10 s. Data from this experiment were collected from all animals before half the cohort ($n = 4$) underwent surgery. Two weeks after the second stage of surgery, or an equivalent period of rest for the control group, we tested all monkeys for two 1000-trial sessions using the same stimuli and then two additional 1000-trial sessions using a novel set of two stimuli to ensure that any changes to their behavior was not attributed to any built-up association with the preoperatively used stimuli.

**Data analysis.** Trials were split into match and nonmatch trials and then into correct and incorrect responses. Each trial was ranked by latency to respond and grouped into ten equally sized bins to ensure that each monkey was contributing equally to each bin. For each response-speed decile, we thus had a measure of false alarms, misses, hits, correct rejections, and overall accuracy as measured by d'. We compared overall accuracy and patterns of different error types via mixed ANOVA with $\alpha = 0.05$ and estimated effect size as partial eta squared. To better discriminate between null and alternative hypotheses, we also ran a Bayesian analysis using an uninformative prior on the critical comparison of false alarm rates at the quickest response decile from before to after surgery or rest. We report the Bayes Factor ($BF_{01}$), which represents the ratio of evidence in favor of the null hypothesis relative to the alternative hypothesis (i.e., 2 = twice as much evidence in favor of the null hypothesis, 1 = equal evidence in favor of both hypotheses, 0.5 = twice as much evidence in favor of the alternative hypothesis).

## Experiment 2

**Testing procedure.** Monkeys completed two 600-trial sessions of a modified yes/no recognition task (Fig. 3a). Monkeys initiated a trial by touching a green start box at the bottom of the screen (100 × 100 pixels), saw a sample image (300 × 300 pixels), and then touched it to progress. Stimuli were color clipart images similar to those in Experiment 1. After a 4-s retention interval (first session) or 20-s retention interval (second session), monkeys were presented with one test image and a nonmatch symbol, as in Experiment 1. Reward, timeout, and ITI contingencies were as described in Experiment 1. The first 100 trials of each session were normal trials used for warm-up and were not analyzed. For the following 500 trials, ten percent of the trials were probe nonmatch trials in which the to-be-rejected test image was the same as the sample and target image on the match trial immediately prior (see Fig. 3a). Thus, these trials appeared infrequently, at a 10% density over the last 500 trials and 8.3% density over the whole session, and should be probing steady-state behavior. Reward contingencies in probe trials were normal. All images were trial-unique with the exception of the probe trials which had been seen on the previous trial. Randomization of trial type and screen location was as in Experiment 1. We collected two sessions (one with 4-s retention intervals and one with 20-s retention intervals) preoperatively just after acquisition of the preoperative data from Experiment 1, and then two additional sessions (one with 4-s retention intervals and one with 20-s retention intervals) postoperatively just after acquisition of the postoperative data from Experiment 1.

**Data analysis.** For each session, false alarms were calculated for the 200 normal baseline nonmatch trials in the latter part of the session and the 50 probe nonmatch trials. Differences were explored using mixed ANOVA (between-subjects factor of lesion group and within-subjects factors of surgical timepoint, trial type, and within-trial interval length) and estimated effect size as partial eta squared. We conducted planned comparisons with paired t tests and estimated effect size as Cohen's d. For all tests, $\alpha = 0.05$. To better discriminate between null and alternative hypotheses, we also ran a Bayesian analysis using an uninformative prior on the critical comparison of false alarms to the recently seen probe lures. We report the Bayes Factor ($BF_{01}$), which represents the ratio of evidence in favor of the null hypothesis relative to the alternative hypothesis (i.e., 2 = twice as much evidence in favor of the null hypothesis, 1 = equal evidence in favor of both hypotheses, 0.5 = twice as much evidence in favor of the alternative hypothesis).

## Experiment 3

**Training procedure.** We first accommodated monkeys to the shifting payout structure that would produce the different criteria levels.

Monkeys started by completing 200 trials per day of a yes/no recognition task with trial-unique stimuli, similar to Experiment 2. Stimuli were composed of color clipart images (300 × 300 pixels) and were the same as used in the prior ROC study with monkeys[26]. Reward contingencies were similar to Experiment 2 with two exceptions: (a) to ensure that all bias came from the amount of food rather than the type of food, all rewards were now grain-based pellets; and (b) amount of food earned per correct answer varied and averaged three rewards per correct response (Fig. 4a). To ensure that monkeys maintained motivation while earning multiple pellets per correct response, we increased the number of pellets per correct responses from one to two to three in stages. Monkeys progressed to the next stage of training when they completed three consecutive sessions at or above 80% accuracy. Training was complete when monkeys finished three consecutive sessions earning three food pellets per correct response at or above 80% accuracy. In addition, we monitored response bias levels to ensure monkeys did not develop a bias during training. Data from this training set were not used in the experiment.

**Testing procedure.** Monkeys completed 25 200-trial sessions of the trained yes/no recognition task in which the relative reward for hits and correct rejections varied day-to-day (Fig. 4a). As in Experiment 1, rewarded trials were considered hits and correct rejections. The retention delay was 20 s and the ITI was 5 s. To shift the monkey's decision criteria, we used five different bias levels in which the ratio of rewards for correct rejections to hits was: 5:1, 4:2, 3:3, 2:4, and 1:5 (Fig. 4a). The order of bias levels was pseudo-randomly distributed across sessions and varied per monkey. All monkeys experienced each bias level five times. Under this payout structure, the optimal response is always to remember and to answer correctly. However, when uncertain of his memory, the monkey might shift his decision criterion in favor of the biased rewards, accepting the test item more often when hits are heavily rewarded and rejecting it when correct rejections are heavily rewarded. In this way, the reward bias functions like the different difficulty levels used with rodents[7,19] and confidence ratings used with humans[16,28].

**Data analysis.** Because bias levels changed each day and the monkeys had to learn the current bias level through trial and error, we discarded the initial 20% of trials for each session. As a manipulation check after all data were collected, we evaluated each session to determine whether the bias manipulation had appropriately shifted each monkey's criterion parameter (c'). For a minority of sessions (median = 7% of data), monkeys did not appropriately shift their criterion in response to the reward bias, as evidenced by a criterion parameter that was not numerically between the criterion parameters of the neighboring bias levels. Consequently, we analyzed the ROC curves with those outlier sessions discarded. We calculated all relevant signal detection measures[65] as a function of response bias for each monkey. We then used the dual process analysis spreadsheet provided online by Yonelinas (https://yonelinas.faculty.ucdavis.edu/roc-analysis/) to derive ROC curves and parameter estimates of recollection and familiarity for each monkey. We compared bias levels via mixed ANOVA (between-subjects factor of group and within-subjects factor of bias level), estimating effect size as partial eta squared, and compared parameter estimates via independent samples t-tests, both with $\alpha = 0.05$. To better discriminate between null and alternative hypotheses, we also ran a Bayesian analysis using an uninformative prior on the critical parameter estimates of recollection and familiarity. We report the Bayes Factor ($BF_{01}$), which represents the ratio of evidence in favor of the null hypothesis relative to the alternative hypothesis (i.e., 2 = twice as much evidence in favor of the null hypothesis, 1 = equal evidence in favor of both hypotheses, 0.5 = twice as much evidence in favor of the alternative hypothesis).

## Experiment 4

**Training procedure.** Training proceeded in a manner similar to that described by Browning et al.[30]. Monkeys initiated a trial by touching a green start box at the bottom of the screen (100 × 100 pixels) and then saw images (300 × 300 pixels) on the left and right side of the screen. Stimuli were color clipart images similar to those in Experiment 1. On each trial, one was pre-designated as the correct response (S+) and the other as the incorrect response (S−). Touching the S+ image resulted in the delivery of one food reward pellet, a secondary audio reinforcer ("excellent!" or "woo-hoo!"), and both images remained on the screen for an additional 500 ms. Touching the S− image resulted in no food reward, a secondary audio signal ("d'oh!"), and both images disappeared immediately. In addition, the screen location and presentation order of the S− stimuli within a day were pseudo-randomized. There was a 10-s ITI following a correct response or 20-s ITI following an incorrect response. Any touches to the screen during the ITI reset the interval. On day 1, the assignment of S+ and S− within each image pair was random and monkeys had to guess. On subsequent days, the images could be discriminated on the basis that the S+ was novel and the S− had been seen on previous days (Fig. 5a). To teach monkeys this rule, they first learned one training set consisting of 100 problem pairs. Initially, they learned 25 pairs until they were selecting the S+ on more than 90% of trials for two consecutive days. Then 25 additional problem pairs were added (50 pairs total) and monkeys tested until they reached the same criterion. Finally, 50 additional problem pairs were added (100 pairs total). Monkeys completed one session per day. Training was complete when a monkey performed at or above 90% accuracy for two consecutive days on the full set of 100 problem pairs. Data from this training set was not used in the experiment.

**Testing procedure.** We tested monkeys on three 100-pair sets in sequence. Monkeys completed one 100-trial session per day. A different pair of images appeared on each trial. On day 1 with each set, all images in the set were novel; on subsequent days, the S+ images were novel and the S− images were the same as in all previous days with that set (Fig. 5a). Each image set was used until the monkeys learned to discriminate between novel and familiar items with 90% accuracy or greater for two consecutive days, at which point monkeys were trained on the next image set.

**Data analysis.** We compared sessions to criterion separately for each of the three problem sets using independent samples t-tests with α = 0.05. To better discriminate between null and alternative hypotheses, we also ran a Bayesian analysis using an uninformative prior on the critical data of sessions to criterion. We report the Bayes Factor (BF$_{01}$), which represents the ratio of evidence in favor of the null hypothesis relative to the alternative hypothesis (i.e., 2 = twice as much evidence in favor of the null hypothesis, 1 = equal evidence in favor of both hypotheses, 0.5 = twice as much evidence in favor of the alternative hypothesis). We also compared the slope of each monkey's learning with a mixed ANOVA with a between-subjects factor of group (amygdala or control) and a within-subjects factor of problem set (1, 2, or 3) and estimated effect size as partial eta squared.

## Experiment 5

**Training procedure.** We trained monkeys on a touchscreen variant of a previously used multi-arm bandit reinforcement learning task[33]. On each trial, the monkey started the trial by touching a central green square (100 × 100 pixels). Three color photographs (143 × 190 pixels) appeared in three of six possible locations equidistant from the center of the screen in either an upright or inverted triangular pattern (Fig. 6a). Photographs were the same as used previously[33] and were normalized for mean luminance and spatial frequency. Each

image was initially novel and pseudo-randomly assigned a high, medium, or low probability of reward, with the only constraint that all three images could not have the same probability. Selecting an image delivered a single nutritionally complete food pellet at the associated probability and, if reward was delivered, a secondary audio reinforcer ("excellent!" or "woo-hoo!"). Failure to select an image within 3 s resulted in a different audio cue ("d'oh") and aborted the trial. Aborted trials were not repeated. All trials were separated by a 2-s ITI. Occasionally, one of the images was replaced with a novel image and pseudo-randomly assigned to a high, medium, or low reward probability. During initial training, the reward probabilities were 90%, 50%, and 10% and a novel image appeared every 30−50 trials. Monkeys completed between 800 and 1200 trials/day depending on individual motivation. Once monkeys were selecting the best available image significantly more frequently than the worst available image, as measured by a chi-square test, on two consecutive days, they proceeded to testing.

**Testing procedure.** For critical testing, the high, medium, and low reward probabilities were 75%, 50%, and 25%. Novel images appeared every 10−30 trials with the exception that the first image replacement of the session had to occur after at least 15 trials. Monkeys completed 800 to 1200 trials/day, depending on individual motivation, until they had completed 12,000 total trials, which took a mean of 11.5 days.

**Data analysis.** We quantified learning in two ways. First, we examined how frequently the monkeys selected novel choice options over the first twenty trials since they were introduced. During this epoch the monkeys had the opportunity to learn whether an option was assigned a high, medium, or low reward probability. We used a sliding t-test to compare how frequently options with different reward rates were selected from one trial to the next, as the monkeys sampled the novel options. To capture both fragile and robust learning, we present those results both uncorrected and corrected for multiple comparisons. Second, we modeled the learning of each animal using the same reinforcement learning (RL) model previously used with this task[33,54].

**Reinforcement learning model.** The RL model is explained in detail elsewhere[33,54], but briefly, we fit a RL model to the choice behavior of the monkeys to estimate learning rates, inverse temperature, and the value of novel stimuli. The model was fit separately to the choice behavior from each block of trials within a session. The model updates the value, $v$, of a chosen option, $i$, based on reward feedback, $r$ in trial $t$ as:

$$v_i(t) = v_i(t-1) + \alpha\left(r(t) - v_i(t-1)\right) \tag{1}$$

Thus, the updated value of an option is given by its old value, $v_i(t{-}1)$ plus a change based on the reward prediction error ($r(t){-}v_i(t{-}1)$), multiplied by the learning rate parameter, $\alpha$. When a novel stimulus is introduced in trial $t'$, there is no reward history. The value of an option when it was first introduced was fit as a free parameter in the model, $v_i(t') = v_i^0$, where $t'$ is the first trial for a novel option. Thus, whenever we introduced a novel option, we substituted $v_i^0$, into the model, and this was the value which was updated on subsequent trials following feedback. The relative propensity of the monkeys to pick the novel option when it was introduced allowed us to estimate the value of that option relative to the other available options. The more often they picked the novel option when it was introduced, the higher the value of novel options. This is particularly true if the novel option is chosen when the other available options are of high value. The free parameters (the initial value of novel options, $v_i^0$, the learning rate parameter, $\alpha$ and the inverse temperature, $\beta$, which estimates how consistently animals choose the highest valued option), were fit by maximizing the

likelihood of the choice behavior of the participants, given the model parameters. Specifically, we calculated the choice probability $d_i(t)$ using:

$$d_i(t) = \frac{\exp(\beta v_i(t))}{\sum_{k=1}^{3} \exp(\beta v_k(t))} \qquad (2)$$

And then calculated the log-likelihood as:

$$ll = \sum_{t=1}^{T} \log \sum_{k=1}^{3} c_k(t) d_k(t) \qquad (3)$$

Where $c_k(t) = 1$ when the subject chooses option $k$ in trial $t$ and $c_k(t) = 0$ for all unchosen options. In other words, the model maximizes the choice probability ($d_k(t)$) of the actual choices the participants made. $T$ is the total number of trials in the block for each monkey. To avoid local minima, initial value and learning rate parameters were chosen randomly, drawn from a normal distribution with a mean of 0.5 and a standard deviation of 3. The inverse temperature parameter was randomly drawn from a normal distribution with a mean of 1 and a standard deviation of 5. No constraints were placed on the estimated parameters. Model fits were repeated 10 times to avoid local minima and the fit with the minimum log-likelihood was selected as the best fit. Likelihood ratio tests were used to compare the fit between the estimated model and a null model that assigned novel options a fixed value of 0.5 (e.g., their empirical average reward expectation).

The result of this model is the estimation of three key free parameters that characterize each monkey's learning. First, the learning rate, $\alpha$, quantifies the speed of the learning. Second, the inverse temperature, $\beta$, quantifies the consistency with which subjects select the most valuable option. Third, the novelty bonus, or penalty, quantifies biases in novelty seeking or avoidance reflecting, in part, how uncertainty is valued (e.g., with no information, a novel item should be valued at 0.5, but a monkey with a novelty penalty might undervalue a novel item at 0.4). During model fitting, we omitted individual testing sessions for which one or more parameter did not converge, which resulted in omitting 3.4% of the total data for this specific analysis. The amount of omitted data did not differ between the two groups ($F(1, 6) = 0.460$, $p = 0.523$). The effect of lesion status on these parameters was then tested with a linear mixed-effects model in which session and monkey were random effects, with session nested within monkey and monkey nested within groups, and group was a fixed effect. We estimated effect size as partial eta squared. A deficit in reward learning would manifest as either a lower learning rate or a lower inverse temperature, whereas a deficit in familiarity might manifest as an atypical novelty bonus/penalty. To better discriminate between null and alternative hypotheses, we also ran a Bayesian analysis using an uninformative prior on the critical parameter estimates of learning rate, inverse temperature, and novelty bonus. We report the Bayes Factor ($BF_{01}$), which represents the ratio of evidence in favor of the null hypothesis relative to the alternative hypothesis (i.e., 2 = twice as much evidence in favor of the null hypothesis, 1 = equal evidence in favor of both hypotheses, 0.5 = twice as much evidence in favor of the alternative hypothesis).

### Reporting summary
Further information on research design is available in the Nature Portfolio Reporting Summary linked to this article.

## Data availability
Data for all figures in this paper are provided in the accompanying Source data file and the manuscript tables. Other data generated during and/or analyzed during the current study are available from the corresponding author on request. Source data are provided with this paper.

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

## Acknowledgements

We thank Andrew R. Mitz and Jaewon Hwang for their help, advice, and technical support. We thank Philip G. F. Browning, Sebastian Guderian, and Alison R. Weiss for methodological advice and access to stimuli and unpublished data. We thank Emily Moylan, Dawn Anuszkiewicz-Lundgren, and Richard Saunders for their help with MR scanning. We thank

Justin Golomb, Jim Fellows, and Krystal Allen-Worthington for surgical and veterinary assistance. This work was supported by the Intramural Research Program of the National Institute of Mental Health (ZIA MH002736).

## Author contributions

B.M.B and E.A.M. conceived of the study. B.M.B., J.L.S., C.L.K. and D.R.L. collected the data. B.M.B., E.A.M., and J.L.S. performed the surgeries. B.M.B. and V.D.C. analyzed the data. B.M.B., V.D.C. and E.A.M. wrote the manuscript. E.A.M. supervised all aspects of the study. All authors discussed the results, helped form the interpretation, and commented on the manuscript at all stages.

## Competing interests

The authors declare no competing interests.
