## [Peer Review File · Nature Communications]

REVIEWER COMMENTS

Reviewer #1 (Remarks to the Author):

The role of the amygdala in recognition memory has been controversial. Studies involving human patients or monkeys with amygdala damage have produced mixed findings. Some found that these individuals showed impairment in the familiarity process, while others found an absence of such impairment. The current study addresses this issue by performing five experiments involving eight monkeys, half of which received an excitotoxic lesion targeted at the amygdala. Three experiments consistently showed no familiarity impairment after the amygdala lesion, and a fourth experiment (Experiment 2) showed an impairment. The authors then ran a fifth experiment and suggested that the odd finding in Experiment 2 could be explained by anomalies in reward processing. Overall, this is an interesting study that addresses the role of the amygdala in the familiarity process. It is appealing that the authors performed five experiments, with the last experiment attempting to explain odd findings. I am particularly keen on having a better understanding of the data of the last experiment.

Comment 1. Could the intended amygdala lesion be extended to the entorhinal cortex and perirhinal cortex? I suggest that Table 1 could be adapted to show whether there was any damage in these two regions that are also related to memory processes.

Comment 2. The key argument that familiarity in memory does not require amygdala was heavily based on frequentist statistics, which is not ideal for supporting null findings. I suggest the authors using Bayesian statistics to support their null findings.

Comment 3. Please include a table showing the results of the ANOVA associated with Figure 3B.

Comment 4. Lines 333-334: "Notably, both the control and amygdala lesions groups showed an initial avoidance of novel choice options and appeared to explore them at a similar rate". Which part of the data shows that the monkeys initially avoided novel options?

Comment 5. It would be useful to show how monkeys responded to the novel option when it was first introduced. Currently, Figure 6c only shows the responses as a function of the reward probabilities.

Comment 6. Did the control monkey receive any surgical procedures? Please describe that in the Surgery section.

Comment 7. Please show the equation(s) of the reinforcement learning model in the Methods.

Comment 8. The author hoped to claim that the amygdala lesion did not cause any familiarity impairment, based on results of Experiments 1, 3, and 4. They attempted to explain the odd findings in Experiment 2 (i.e. after amygdala lesion, there was a reduction in false alarms to highly familiar probe lures) by performing Experiment 5. However, I have concerns whether the current results of Experiment 5 are sufficient to explain the odd results of Experiment 2:

Comment 8a. First, based on the results of Figure 6b, I am not sure if it supports the title in Line 299. Figure 6b shows that the inverse temperature parameter, which reflects choice consistency, is smaller in amygdala lesion monkeys. This suggests that they were, in general, more random in their choices. So, I think claiming that it is a “reward processing” problem could potentially cause confusions to some readers – “reward processing” sounds more related to the learning rate parameter in the authors’ reinforcement learning model.

Comment 8b. Second, if amygdala lesion monkeys were more random in their choices in general, I would expect that they would also show more false alarms in all types of stimuli in Experiment 2. However, this was not the case. Any explanations to this?

Comment 8c. Third, I agree that the results of Experiment 2 and 5 may be explained by reward processing or reinforcement learning problem. Previous studies have demonstrated that there are possibly two modes of learning. First, a contingent learning mechanism that learns precisely the association between an object and the reward that is causally related to the object. Second, a non-contingent learning mechanism that draws an association between an object and a reward, as long as they occur relatively close in time even if they do not have a causal relationship. It has been shown that the amygdala is involved in non-contingent learning (e.g. Jocham et al., *Neuron*, 2016; Chau et al., *Neuron*, 2015). I wonder if the data in Experiment 5 could be better explained by a model that captures non-contingent learning? If a non-contingent learning parameter was poorer in the amygdala lesion monkeys, it may help explaining why they showed less false alarms in Experiment 2 by arguing that the error-prone non-contingent learning mechanism was knocked out.

Reviewer #2 (Remarks to the Author):

The study examines the role of the amygdala in supporting familiarity and recollection based episodic memory discriminations, by examining the effects of bilateral amygdala lesions on recognition memory in rhesus monkeys. There is currently considerable debate over this issue because some previous work suggests it plays critical role (e.g., a rodent study by Farovik et al., 2011), whereas other work indicates that it does not play a critical role (e.g., human lesion studies). The current study examines the role of the amygdala using four different behavioral studies (i.e., tendencies to make low-latency false alarms, to make false alarms to recently-seen lures, to produce curvilinear ROC curves, and to discriminate stimuli based on repetition across days). Three of the 4 methods indicate that familiarity is not impaired

by amygdala lesions. The fourth method suggested that familiarity may be disrupted, but a subsequent experiment indicated that this was explained by a deficit in reward processing rather than a familiarity impairment.

The study addresses an important theoretical debate in the literature and so provides important new results regarding the role of the amygdala in recognition memory. The lesion methods and the various behavioral methods are well controlled and well suited to address the specific aim. Moreover, the results across the various methods in general converge in providing a clear picture of the role of the amygdala in recognition. The one anomalous result was found to be well explained as reflecting a reward processing problem rather than a familiarity deficit. Thus, the conclusions are well supported by the results. So I think the study will make an important contribution to the literature.

However, there were several issues that should be addressed.

First, there are a number of human episodic memory studies that have examined the effects of amygdala lesions on episodic memory that should be integrated with the current results. Namely, there are a number of studies that have indicated that amygdala lesions do not generally impair episodic memory in humans, except in the case when the materials are emotional in nature (e.g., Adolphs et al., 1995; Cahill et al., 1995; Markowitsch et al., 1994). These results would seem to be in agreement with the current findings that familiarity is not impaired by amygdala lesions.

Second, I felt that the one published finding that does not fit with the current results (Farovik et al., 2011 that used the ROC method with rodents and concluded that familiarity was disrupted), could be considered in more detail. The authors are correct in pointing out that this discrepancy could be due to anatomical differences or methodological differences across studies. However, there do seem to be several other possibilities that should be discussed. For example, another possibility is that it reflects a false positive finding, and thus may not have held up in studies that utilized multiple convergent methods. In addition, I wondered if there is any existing evidence suggesting that the methodological difference may be important? For example, are there studies that suggest that rats and monkeys respond differently to reward manipulations? Or that reward learning effects differ at all when differences in reward are supplemented with differences in difficulty (as was done in the Farovik et al., study?)

Third, I think more detail could be provided about the specific methods used in the various experiments. For example, the ROC and the reward learning modelling methods are probably not very familiar to many readers and so additional info could have been provided.

Reviewer #3 (Remarks to the Author):

I reviewed the manuscript entitled « The amygdala is not necessary for the familiarity aspect of recognition memory”.

I found the paper very convincing; the problem was well described with a concise but relevant theoretical introduction, and a solid methodology across 5 experiments in amygdala-lesioned, and control counterparts, non-human primates.

Three out of 4 experiments that tested familiarity using different manipulations showed results of preserved familiarity in monkeys with amygdala lesions, while one experiment showed impaired familiarity, but this result was interpreted in terms of reward processing impairment, this interpretation being supported by the results from a 5th experiment assessing reward processing specifically and showing deficits in amygdala-lesioned monkeys.

I only have one major concern about this manuscript, which concerns this question of reward processing impairment in amygdala-lesioned monkeys. Exp.5 indeed shows deficit in reward processing in these monkeys, with, apparently, a difficulty in discriminating between stimuli that predicted the medium and high reward probability. Based on this result, and on the discrepant result of Exp.2, the authors propose an interpretation of the familiarity impairment seen in Exp.2 in terms of deficit in processing the reward associated with a given stimulus rather than familiarity impairment itself.

What I wonder is, if this is true in Exp.2 and 5, shouldn't this be true in other experiments that also involve reward processing? Notably, Exp.3 manipulates response bias by manipulating the reward value. If monkeys with amygdala lesions display deficit in reward processing, how come that their decision criterion was affected by reward manipulation to a similar extent than in controls? Similarly, in Exp. 4, monkeys learn the correct response S+ through reward manipulation, and there too, monkeys with amygdala lesion do not display deficits. I think it would be good to tackle this question.

Finally, I have a minor comment concerning Exp. 1, because the text from the introduction of this specific experiment seems to suggest that there's going to be a response deadline, while, from what I gathered, the analyses focused on the speed/the time course of false alarms without a response deadline. I think this could be clearer.

Response to reviews for NCOMMS-23-00867-T

Please find below our responses interleaved with the full text of the action letter. We especially thank the reviewers for their thorough, thoughtful, and extremely helpful feedback. As requested, we have indicated major changes within the manuscript with **colored text**.

REVIEWER COMMENTS

Reviewer #1 (Remarks to the Author):

The role of the amygdala in recognition memory has been controversial. Studies involving human patients or monkeys with amygdala damage have produced mixed findings. Some found that these individuals showed impairment in the familiarity process, while others found an absence of such impairment. The current study addresses this issue by performing five experiments involving eight monkeys, half of which received an excitotoxic lesion targeted at the amygdala. Three experiments consistently showed no familiarity impairment after the amygdala lesion, and a fourth experiment (Experiment 2) showed an impairment. The authors then ran a fifth experiment and suggested that the odd finding in Experiment 2 could be explained by anomalies in reward processing. Overall, this is an interesting study that addresses the role of the amygdala in the familiarity process. It is appealing that the authors performed five experiments, with the last experiment attempting to explain odd findings. I am particularly keen on having a better understanding of the data of the last experiment.

Thank you. We're glad you found the study interesting.

Comment 1. Could the intended amygdala lesion be extended to the entorhinal cortex and perirhinal cortex? I suggest that Table 1 could be adapted to show whether there was any damage in these two regions that are also related to memory processes.

These data are presented in text (lines 606-614). We present the group means of hypersignal coverage and estimated damage (with uncertainty ranges) for both the entorhinal cortex and the perirhinal cortex, as well as the hippocampus. The relevant section reads:

Total unintended damage was generally low across surrounding structures and usually unilateral. Based on regression functions described previously⁶⁰, unintended damage was estimated as follows: hippocampus: hypersignal coverage = 9.7%, estimated damage = 4.0% (95%PI: 0.0%-11.9%); entorhinal cortex: hypersignal coverage = 15.0%, estimated damage = 5.7% (95%PI: 0.0%-20.0%); perirhinal cortex: hypersignal coverage = 8.9%, estimated damage = 3.0% (95%PI: 0.1%-7.8%;). Thus, the combination of substantial bilateral damage to the amygdala and minor, unilateral damage to surrounding

structures should produce a valid test of the necessity of the amygdala for these tasks.

We also now further highlight the relevance of this to prior studies in the general discussion (lines 410-413). The relevant section reads:

In our study, the relative sparing of the perirhinal cortex (estimated damage = 3.0%) and lack of robust familiarity deficit is consistent with the hypothesis that Patient NB's familiarity deficit might be due to her perirhinal damage^{16,17}.

Comment 2. The key argument that familiarity in memory does not require amygdala was heavily based on frequentist statistics, which is not ideal for supporting null findings. I suggest the authors using Bayesian statistics to support their null findings.

As suggested, we now include a Bayes Factor (BF_{01}) for each of our critical test as a measure of how much more that evidence supports the null hypothesis than the alternative hypothesis.

Comment 3. Please include a table showing the results of the ANOVA associated with Figure 3B.

As suggested, we now present the full ANOVA table as Table S1 in the supplemental materials.

Comment 4. Lines 333-334: "Notably, both the control and amygdala lesions groups showed an initial avoidance of novel choice options and appeared to explore them at a similar rate". Which part of the data shows that the monkeys initially avoided novel options?

The initial avoidance of novel items is shown in the leftmost points of Figure 6c, which depicts choice of the novel item as a function of trials since the introduction of the novel item. To make this clearer, the revised text (lines 341-343) now reads:

Notably, both the control and amygdala lesion groups showed an initial avoidance of novel choice options (Figure 6c, leftmost points) and appeared to explore them at a similar rate.

Comment 5. It would be useful to show how monkeys responded to the novel option when it was first introduced. Currently, Figure 6c only shows the responses as a function of the reward probabilities.

Figure 6c does show how monkeys responded to the novel option when it was first introduced. For each group, the leftmost point is the trial on which the novel option is first introduced (and

thus truly novel), the second point from the left is the second trial on which the novel option is present (and thus has been seen exactly once before), etc. The relevant lines (lines 371-372) from the figure legend read:

Proportion (\pm SEM) of times a novel option was chosen as a function of assigned reward probability (75%, 50%, or 25%) and number of trials since the introduction of a novel item. Trial 0 is the trial on which the novel item was introduced.

Comment 6. Did the control monkey receive any surgical procedures? Please describe that in the Surgery section.

As suggested, we now describe the control monkeys' experience in more detail (lines 589-592). The relevant section reads:

Control monkeys were unoperated. Each control monkey was randomly matched with one of the operated monkeys and rested a number of days equal to the total time between that monkey's last preoperative testing day and first postoperative testing day (mean = 51 days). During this interval, there was no cognitive testing; food and water were available ad libitum.

Comment 7. Please show the equation(s) of the reinforcement learning model in the Methods.

As suggested, we now show the equations of the reinforcement learning model in the Methods (lines 823-857). The relevant lines read:

Reinforcement learning model

The RL model is explained in detail elsewhere^{33,54}, but briefly, we fit a RL model to the choice behavior of the monkeys to estimate learning rates, inverse temperature, and the value of novel stimuli. The model was fit separately to the choice behavior from each block of trials within a session. The model updates the value, v , of a chosen option, i , based on reward feedback, r in trial t as:

$$v_i(t) = v_i(t-1) + \alpha(r(t) - v_i(t-1))$$

Thus, the updated value of an option is given by its old value, $v_i(t-1)$ plus a change based on the reward prediction error ($r(t)-v_i(t-1)$), multiplied by the learning rate parameter, α . When a novel stimulus is introduced in trial t' , there is no reward history. The value of an option when it was first introduced was fit as a free parameter in the model, $v_i(t')=v_i^0$, where t' is the first trial for a novel option. Thus, whenever we introduced a novel option, we substituted v_i^0 , into the model, and this was the value which was updated on subsequent trials following feedback. The relative propensity of the monkeys to pick the novel option when it was introduced allowed us to estimate the value of that option relative to the other available options. The more often they picked the novel option when it was introduced, the higher the value of novel options. This is

particularly true if the novel option is chosen when the other available options are of high value. The free parameters (the initial value of novel options, v_i^0 , the learning rate parameter, α and the inverse temperature, β , which estimates how consistently animals choose the highest valued option), were fit by maximizing the likelihood of the choice behavior of the participants, given the model parameters. Specifically, we calculated the choice probability $d_i(t)$ using:

$$d_i(t) = \frac{\exp(\beta v_i(t))}{\sum_{k=1}^3 \exp(\beta v_k(t))}$$

And then calculated the log-likelihood as:

$$\# = \sum_{t=1}^T \log \sum_{k=1}^3 c_k(t) d_k(t)$$

Where $c_k(t)=1$ when the subject chooses option k in trial t and $c_k(t)=0$ for all unchosen options. In other words, the model maximizes the choice probability ($d_k(t)$) of the actual choices the participants made. T is the total number of trials in the block for each monkey. To avoid local minima, initial value and learning rate parameters were chosen randomly, drawn from a normal distribution with a mean of 0.5 and a standard deviation of 3. The inverse temperature parameter was randomly drawn from a normal distribution with a mean of 1 and a standard deviation of 5. No constraints were placed on the estimated parameters. Model fits were repeated 10 times to avoid local minima and the fit with the minimum log-likelihood was selected as the best fit. Likelihood ratio tests were used to compare the fit between the estimated model and a null model that assigned novel options a fixed value of 0.5 (e.g., their empirical average reward expectation).

Comment 8. The author hoped to claim that the amygdala lesion did not cause any familiarity impairment, based on results of Experiments 1, 3, and 4. They attempted to explain the odd findings in Experiment 2 (i.e. after amygdala lesion, there was a reduction in false alarms to highly familiar probe lures) by performing Experiment 5. However, I have concerns whether the current results of Experiment 5 are sufficient to explain the odd results of Experiment 2:

Comment 8a. First, based on the results of Figure 6b, I am not sure if it supports the title in Line 299. Figure 6b shows that the inverse temperature parameter, which reflects choice consistency, is smaller in amygdala lesion monkeys. This suggests that they were, in general, more random in their choices. So, I think claiming that it is a “reward processing” problem could potentially cause confusions to some readers – “reward processing” sounds more related to the learning rate parameter in the authors’ reinforcement learning model.

Here, we use “reward processing” as a catch-all term for the multiple processes that result in the ability to base current actions on past rewards. To make this more explicit, we have added the following language (lines 321-322):

Such an impairment in reward processing might be due to an impairment in the speed of reward learning, the consistency of reward-guided choices, or both.

Comment 8b. Second, if amygdala lesion monkeys were more random in their choices in general, I would expect that they would also show more false alarms in all types of stimuli in Experiment 2. However, this was not the case. Any explanations to this?

Please see the next comment, which addresses both these two points.

Comment 8c. Third, I agree that the results of Experiment 2 and 5 may be explained by reward processing or reinforcement learning problem. Previous studies have demonstrated that there are possibly two modes of learning. First, a contingent learning mechanism that learns precisely the association between an object and the reward that is causally related to the object. Second, a non-contingent learning mechanism that draws an association between an object and a reward, as long as they occur relatively close in time even if they do not have a causal relationship. It has been shown that the amygdala is involved in non-contingent learning (e.g. Jocham et al., *Neuron*, 2016; Chau et al., *Neuron*, 2015). I wonder if the data in Experiment 5 could be better explained by a model that captures non-contingent learning? If a non-contingent learning parameter was poorer in the amygdala lesion monkeys, it may help explaining why they showed less false alarms in Experiment 2 by arguing that the error-prone non-contingent learning mechanism was knocked out.

We agree that it is informative to consider our data through the lens of the proposed role for the amygdala in non-contingent learning. Indeed, this view might reconcile the seemingly inconsistency you raise in Comment 8b. The added discussion (lines 490-522) now reads:

One potential explanation for why a deficit in reward processing affected Experiments 2 and 5, but not Experiments 1, 3, and 4, might lie in the amygdala’s proposed role in noncontingent statistical learning^{56,57}. When learning about rewards, organisms learn not only the causal contingencies of which action/choice produced a reward, but also statistical regularities about noncontingent actions/choices/cues that occurred in the penumbra of the reward. For example, when you select a specific image and are rewarded, you learn that choice/reward contingency, but that association might spread to the nearby, noncontingent things such as your choice on the previous trial. When an organism completes the same action repeatedly, contingent and noncontingent learning are usually consistent. But when contingencies shift, when associations are still fragile during initial learning, or when one must learn about multiple probabilistic outcomes, contingent and noncontingent learning can be inconsistent. In humans and monkeys, activity in the amygdala

has been found to track this type of noncontingent learning ^{56,57}. In Experiment 5, monkeys needed to rapidly learn the statistical regularities of multiple probabilistically rewarded choices while associations were still fragile and uncertain, so the influence of noncontingent learning on behavior was likely high. Removing this noncontingent learning about statistical regularities might have resulted in irregular choices, manifesting in the observed lower choice consistency. In Experiment 2, for which images were also relatively novel and so associations and visual memory traces were still fragile, the to-be-rejected probe lure was irrelevant to the current trial and actually inconsistent with the memory trace from the current trial. Removing this inconsistent reward association would improve performance, manifesting in the observed lower error rate after surgery. In Experiment 1, all targets and lures were highly familiar and had likely acquired similar histories of noncontingent learning, limiting the influence of noncontingent learning on behavior. For Experiment 3, all items were initially novel and items that had acquired both contingent and noncontingent learning in previous trials were never brought back as lures, limiting the influence of noncontingent learning on behavior. For Experiment 4, only the unrewarded items were brought back for subsequent sessions, limiting the influence of noncontingent learning on behavior. In short, the experiments in which we see an effect of amygdala lesions are those in which we might predict the largest role of noncontingent learning in intact animals. Consistent with this idea, a large analysis of monkeys with a variety of focal lesions found that lesioning the amygdala and areas interconnected to it is necessary for using prior beliefs about environmental reward statistics to stabilize value representations when learning in dynamic environments ⁵⁸. Thus, this explanation based on noncontingent learning merits further study.

Reviewer #2 (Remarks to the Author):

The study examines the role of the amygdala in supporting familiarity and recollection based episodic memory discriminations, by examining the effects of bilateral amygdala lesions on recognition memory in rhesus monkeys. There is currently considerable debate over this issue because some previous work suggests it plays critical role (e.g., a rodent study by Farovik et al., 2011), whereas other work indicates that it does not play a critical role (e.g., human lesion studies). The current study examines the role of the amygdala using four different behavioral studies (i.e., tendencies to make low-latency false alarms, to make false alarms to recently-seen lures, to produce curvilinear ROC curves, and to discriminate stimuli based on repetition across days). Three of the 4 methods indicate that familiarity is not impaired by amygdala lesions. The fourth method suggested that familiarity may be disrupted, but a subsequent experiment indicated that this was explained by a deficit in reward processing rather than a familiarity

impairment.

The study addresses an important theoretical debate in the literature and so provides important new results regarding the role of the amygdala in recognition memory. The lesion methods and the various behavioral methods are well controlled and well suited to address the specific aim. Moreover, the results across the various methods in general converge in providing a clear picture of the role of the amygdala in recognition. The one anomalous result was found to be well explained as reflecting a reward processing problem rather than a familiarity deficit. Thus, the conclusions are well supported by the results. So I think the study will make an important contribution to the literature.

Thank you. We are happy that you found the results to be important.

However, there were several issues that should be addressed.

First, there are a number of human episodic memory studies that have examined the effects of amygdala lesions on episodic memory that should be integrated with the current results. Namely, there are a number of studies that have indicated that amygdala lesions do not generally impair episodic memory in humans, except in the case when the materials are emotional in nature (e.g., Adolphs et al., 1995; Cahill et al., 1995; Markowitsch et al., 1994). These results would seem to be in agreement with the current findings that familiarity is not impaired by amygdala lesions.

As suggested, we now expand on the relevance of our current findings to the debate about how to characterize the deficits seen in cases of human amygdala damage, citing the suggested studies (lines 416-423). The current section now reads:

Our study is also consistent with the broader literature on how to characterize the deficits seen in human patients with amygdala damage. Although there are some reports of general memory issues with neutral stimuli in some patients with amygdala damage ^{23,40}, there are also reports in which these patients demonstrate normal memory for neutral stimuli ^{40,41}. Instead, the most robust deficits seem to be in non-mnemonic tasks such as in processing emotions, or in mnemonic tasks that involve emotional stimuli ^{22,41,42}. The current data add to the evidence that amygdala damage is generally not accompanied by general memory impairments.

Second, I felt that the one published finding that does not fit with the current results (Farovik et al., 2011 that used the ROC method with rodents and concluded that familiarity was disrupted), could be considered in more detail. The authors are correct in pointing out that this discrepancy could be due to anatomical differences or methodological differences across studies. However, there do seem to be several other possibilities that should be discussed. For example, another possibility is that it reflects a false positive finding, and thus may not have held up in studies that utilized multiple convergent methods.

We agree that it is possible the finding from Farovik et al. may be a false positive. However, we are hesitant to put too strong an emphasis on this possibility without stronger evidence, such as evaluating the claim using multiple convergent methods in rats, as you suggest. Nevertheless, we agree it is worth mentioning (lines 446-448) The relevant lines read:

A third possibility is that the rodent results ⁴⁹ represent a false positive. However, in the absence of contradictory evidence from a preregistered replication attempt or multiple converging paradigms, we take their results at face value.

In addition, I wondered if there is any existing evidence suggesting that the methodological difference may be important? For example, are there studies that suggest that rats and monkeys respond differently to reward manipulations? Or that reward learning effects differ at all when differences in reward are supplemented with differences in difficulty (as was done in the Farovik et al., study?)

As suggested, we have added citations about how reward and effort are not interchangeable manipulations and about species differences in learning (lines 443-446). The relevant section now reads:

Research demonstrates that manipulations of effort and reward are not interchangeable in their effect on behavior or equivalent in how much they involve the amygdala ⁴⁹. Further, although basic mechanisms of reinforcement learning often seem universal, they actually often differ across species ^{50,51}.

Third, I think more detail could be provided about the specific methods used in the various experiments. For example, the ROC and the reward learning modelling methods are probably not very familiar to many readers and so additional info could have been provided.

As suggested by both you and Reviewer 1, we have expanded the methods sections for the reward learning methods (lines 824-858). Please see our response to their comment for the changed language. For the ROC methods, we are unsure which steps were unclear, but are happy to address any specific issues in future revisions.

Reviewer #3 (Remarks to the Author):

I reviewed the manuscript entitled « The amygdala is not necessary for the familiarity aspect of recognition memory ».

I found the paper very convincing; the problem was well described with a concise but relevant

theoretical introduction, and a solid methodology across 5 experiments in amygdala-lesioned, and control counterparts, non-human primates.

Thank you. We are happy that you found the paper convincing.

Three out of 4 experiments that tested familiarity using different manipulations showed results of preserved familiarity in monkeys with amygdala lesions, while one experiment showed impaired familiarity, but this result was interpreted in terms of reward processing impairment, this interpretation being supported by the results from a 5th experiment assessing reward processing specifically and showing deficits in amygdala-lesioned monkeys.

I only have one major concern about this manuscript, which concerns this question of reward processing impairment in amygdala-lesioned monkeys. Exp.5 indeed shows deficit in reward processing in these monkeys, with, apparently, a difficulty in discriminating between stimuli that predicted the medium and high reward probability. Based on this result, and on the discrepant result of Exp.2, the authors propose an interpretation of the familiarity impairment seen in Exp.2 in terms of deficit in processing the reward associated with a given stimulus rather than familiarity impairment itself.

What I wonder is, if this is true in Exp.2 and 5, shouldn't this be true in other experiments that also involve reward processing? Notably, Exp.3 manipulates response bias by manipulating the reward value. If monkeys with amygdala lesions display deficit in reward processing, how come that their decision criterion was affected by reward manipulation to a similar extent than in controls? Similarly, in Exp. 4, monkeys learn the correct response S+ through reward manipulation, and there too, monkeys with amygdala lesion do not display deficits. I think it would be good to tackle this question.

This seeming inconsistency was also raised by Reviewer 1 in their Comments 8b & 8c. Their interesting suggestion was that it might be explained by looking at our data through the lens of the proposed role of the amygdala in non-contingent learning. The added discussion (lines 490-522) now reads:

One potential explanation for why a deficit in reward processing affected Experiments 2 and 5, but not Experiments 1, 3, and 4, might lie in the amygdala's proposed role in noncontingent statistical learning^{56,57}. When learning about rewards, organisms learn not only the causal contingencies of which action/choice produced a reward, but also statistical regularities about noncontingent actions/choices/cues that occurred in the penumbra of the reward. For example, when you select a specific image and are rewarded, you learn that choice/reward contingency, but that association might spread to the nearby, noncontingent things such as your choice on the previous trial. When an organism completes the same action repeatedly, contingent and noncontingent learning are usually consistent. But when contingencies shift, when associations are still fragile during initial learning, or when one must learn about multiple probabilistic outcomes, contingent and noncontingent learning can be inconsistent. In humans and monkeys, activity in the amygdala

has been found to track this type of noncontingent learning ^{56,57}. In Experiment 5, monkeys needed to rapidly learn the statistical regularities of multiple probabilistically rewarded choices while associations were still fragile and uncertain, so the influence of noncontingent learning on behavior was likely high. Removing this noncontingent learning about statistical regularities might have resulted in irregular choices, manifesting in the observed lower choice consistency. In Experiment 2, for which images were also relatively novel and so associations and visual memory traces were still fragile, the to-be-rejected probe lure was irrelevant to the current trial and actually inconsistent with the memory trace from the current trial. Removing this inconsistent reward association would improve performance, manifesting in the observed lower error rate after surgery. In Experiment 1, all targets and lures were highly familiar and had likely acquired similar histories of noncontingent learning, limiting the influence of noncontingent learning on behavior. For Experiment 3, all items were initially novel and items that had acquired both contingent and noncontingent learning in previous trials were never brought back as lures, limiting the influence of noncontingent learning on behavior. For Experiment 4, only the unrewarded items were brought back for subsequent sessions, limiting the influence of noncontingent learning on behavior. In short, the experiments in which we see an effect of amygdala lesions are those in which we might predict the largest role of noncontingent learning in intact animals. Consistent with this idea, a large analysis of monkeys with a variety of focal lesions found that lesioning the amygdala and areas interconnected to it is necessary for using prior beliefs about environmental reward statistics to stabilize value representations when learning in dynamic environments ⁵⁸. Thus, this explanation based on noncontingent learning merits further study.

Finally, I have a minor comment concerning Exp. 1, because the text from the introduction of this specific experiment seems to suggest that there's going to be a response deadline, while, from what I gathered, the analyses focused on the speed/the time course of false alarms without a response deadline. I think this could be clearer.

As suggested, we have modified the introduction to this experiment to be clearer that this experiment tests the time course (line 134-135). The relevant line now reads:

Here, we tested whether the natural time course of errors – specifically, the elevated false alarms during the quickest responses – was affected by selective amygdala damage.

REVIEWERS' COMMENTS

Reviewer #1 (Remarks to the Author):

I think the manuscript has greatly improved. I only have a few very minor follow-up questions.

Comment 2 continued. The authors have now provided the BF01 for each critical test. Most BF01 scores were below 3 (above which suggests substantial evidence supporting the null hypothesis). Since the null findings were consistent across multiple experiments and animal lesion study often involves small sample size, I think the results remain reasonable. However, I think it is worth mentioning in the Discussion that the relatively small BF01 scores could be a weakness of this study.

Comment 3 continued. According to Table S1, the four way Group \times Timepoint \times Type \times Delay interaction was not significant ($p=0.935$). I think the authors should downplay their claim that the lower false alarms to probe trials in the amygdala group post-op was specific to the long delay condition (around Lines 198-202). I think running posthoc tests for illustrating the significant Group \times Timepoint \times Type interaction would be more important.

Comments 4 and 5 continued. I think I understand now. Figure 6c looks as if it was about the choice probability of the three options, but actually it is all about the novel option only (the reward probability of the novel option can be 25%, 50% or 75%). What about changing the label of the y axis to make this clearer?

Reviewer #2 (Remarks to the Author):

The authors have nicely addressed each of my earlier concerns. I think this will be an important paper.

Reviewer #3 (Remarks to the Author):

The authors have addressed my concerns from the original review

Response to reviews for NCOMMS-23-00867A

Please find below our responses interleaved with the full text of the action letter. We especially thank the reviewers for their thorough, thoughtful, and extremely helpful feedback.

REVIEWERS' COMMENTS

Reviewer #1 (Remarks to the Author):

I think the manuscript has greatly improved. I only have a few very minor follow-up questions.

Comment 2 continued. The authors have now provided the BF01 for each critical test. Most BF01 scores were below 3 (above which suggests substantial evidence supporting the null hypothesis). Since the null findings were consistent across multiple experiments and animal lesion study often involves small sample size, I think the results remain reasonable. However, I think it is worth mentioning in the Discussion that the relatively small BF01 scores could be a weakness of this study.

As suggested, we now include this limitation in the Discussion. The text reads: *“One additional limitation of this study is that our Bayes Factor analyses consistently revealed BF_{01} values under three for null results, which is relatively low support for the null hypothesis compared to the standards of the field. This is likely due to the sample size often used with studies of nonhuman primates. Thus, although the power of our Bayesian analysis is a limitation of this study, it is consistent with the frequentist analysis, consistent across most paradigms, and directly counter to the amygdala hypothesis of familiarity.”*

Comment 3 continued. According to Table S1, the four way Group × Timepoint × Type × Delay interaction was not significant ($p=0.935$). I think the authors should downplay their claim that the lower false alarms to probe trials in the amygdala group post-op was specific to the long delay condition (around Lines 198-202). I think running posthoc tests for illustrating the significant Group × Timepoint × Type interaction would be more important.

As suggested, we have tweaked our wording to remove the language that the effect was not seen in the shorter delay, but instead that it did not reach statistical significance. As for analyzing the Group × Timepoint × Type interaction, we agree that this would normally be our next step if we did not have a planned comparison that was motivated by the experimental design. As it stands, the planned comparison for this design was to compare each group’s performance pre-intervention to their performance post-intervention at both delays. The revised wording now reads *“This change in false alarm rates on probe trials preoperatively to postoperatively was not seen for the control group in any condition and did not reach statistical significance with the amygdala group for the shorter delay”*

Comments 4 and 5 continued. I think I understand now. Figure 6c looks as if it was about the choice probability of the three options, but actually it is all about the novel option only (the reward probability of the novel option can be 25%, 50% or 75%). What about changing the label

of the y axis to make this clearer?

This is a good suggestion. We have changed the Y axis to now read “Proportion choice of novel item”

Reviewer #2 (Remarks to the Author):

The authors have nicely addressed each of my earlier concerns. I think this will be an important paper.

Reviewer #3 (Remarks to the Author):

The authors have addressed my concerns from the original review

We sincerely thank all reviewers for their efforts in strengthening this paper and bringing it to publication.